# Relationship between landslide susceptibility and social lag in Mexico City: The case of the west periphery

**Mario Alejandro Mercado Mendoza**[1]*, **Armando Sánchez Vargas**[2], **Pierre Mokondoko**[3]

**1** Digital Economics Program, Instituto de Investigaciones Económicas, Universidad Nacional Autónoma de México, México City, Mexico, **2** Econometrics Department, Instituto de Investigaciones Económicas, Universidad Nacional Autónoma de México, México City, Mexico, **3** Geoinformatics Department, Instituto de Investigaciones Económicas, Universidad Nacional Autónoma de México, México City, Mexico

☯ These authors contributed equally to this work.
* mario.mercado@ciencias.unam.mx

## Abstract

Landslides threaten sustainable development through economic and human losses. This study integrates machine learning methods to construct susceptibility maps, including topographic-hydrological indicators, to improve the inclusion of earthflow landslides. Furthermore, we aim to find relationships between landslide susceptibility and social lag using Copula models and SHAP values. Results reveal differentiated dependence across different partitions. Specifically, we found regime-specific co-occurrences of high social lag and high landslide susceptibility areas in steep, deprived areas, contrasting resilient affluent zones. Educational deprivation emerges as the top vulnerability factor, followed by healthcare access, overcrowding, and housing deficits. Highlighting spatial inequities, the analysis advocates targeted interventions blending slope stabilization and social policies.

## Introduction

Landslides are a significant natural hazard that can cause human casualties, substantial economic losses, and severe social disruption. In addition, human activity has exacerbated the damage caused by natural hazards and has threatened the sustainable development of cities. Increased urbanization, deforestation, and population growth, particularly in mountainous peri-urban settlements, have often led to unregulated land-use changes that increase the susceptibility of populations and make them highly vulnerable to landslides. To understand and mitigate their impacts, numerous studies have focused on estimating the likelihood of landslide occurrence by developing landslide susceptibility maps [1–4], in which landslide susceptibility is defined as the "likelihood of a landslide occurring in an area based on local terrain conditions" [5].

**Data availability statement:** Data can be found in the following public ZENODO repository:
https://zenodo.org/records/17156313.

**Funding:** The author(s) received no specific funding for this work.

**Competing interests:** The authors have declared that no competing interests exist.

In recent years, an increasing number of robust algorithms and datasets have been developed to enhance the precision of these maps by incorporating factors such as geology, slope, and rainfall patterns, among others. Despite these advances, comparatively fewer efforts have examined how landslide susceptibility levels are associated with social conditions [6]. Although some research has introduced social vulnerability indices and subsequently overlaid them onto susceptibility maps [7,8], few have employed statistical analyses to assess whether higher susceptibility levels are associated with greater social disadvantage. This represents a clear research gap, particularly in rapidly urbanizing regions of emerging economies, where informal settlements often emerge in areas prone to natural hazards [9].

In Mexico, as in many other emerging nations, urban expansion frequently occurs without adequate planning. Specifically, in the western periphery of Mexico City (CDMX), lower-income groups migrating to CDMX often settle on the outskirts of the city due to more affordable housing costs. These populations, having moved from their original communities in search of a better life and higher income, generally seek better opportunities in large cities [10]. Meanwhile, phenomena such as gentrification within the central areas of CDMX have moved established CDMX residents to the periphery [11].

Furthermore, within our study area, in the 1980s a large real estate project known as Santa Fe was launched, aiming to build a "Manhattan" within the city. Developers envisioned the construction of oversized shopping malls, high-end residential areas, and some of the most modern and luxurious office spaces [12]. It was located in a low-income area—possibly because it was less expensive to purchase land there. Currently, this project has expanded and continues to grow to date, along with adjacent luxury zones located in the neighboring Huixquilucan municipality, which has fostered gentrification and exacerbated economic inequality with respect to the original low-income settlers [12].

As a result of all these processes, some of the most vulnerable segments of the CDMX population often end up in the periphery, whether they come from outside or inside the city. This migration pattern is problematic. In particular, in the western periphery of Mexico City, which is a mountainous area with considerable precipitation, the likelihood that lower-income, displaced populations are exposed to landslides increases [13]. Furthermore, such urban growth toward the periphery often lacks essential services such as transportation, reliable water supply, and sewerage systems [14,15].

Generally speaking, Mexico City exemplifies this pattern of vulnerable groups settling in potentially risky areas in the periphery because they lack the income to afford a safer place inside the city, thus heightening their exposure to natural disasters, including landslides [10,11]. This focus is particularly pertinent in Mexico City's western outskirts, where sharp social disparities exist [16], mirroring broader Latin American patterns where impoverished populations frequently inhabit high-risk slopes next to wealthy neighborhoods [9].

Additionally, our study area is mostly classified by Mexico's National Council for the Evaluation of Social Development Policy (CONEVAL) as having low social lag in the AGEB social division, which means that considerably large land areas are

apparently well favored. This contrasts with our on-site observations, in which it is straightforward to find that poor zones are deeply entrenched in the study area, which could lead us to inquire whether CONEVAL's social lag index—because it is not sufficiently granular—may fail to capture the disadvantages that some populations experience. If this is the case, any potential association between landslide susthe ceptibility and CONEVAL social lag index may be weak or spurious.

Given our concern that disadvantaged people may be at additional risk from landslides, our first research question is: is the social lag index positively associated with high landslide susceptibility? Furthermore, given that CONEVAL's index aggregates data over considerably large areas, we also need to analyze each individual variable comprising this social lag index. Thus, our second research question is: which of the social variables that comprise the social lag index—considered individually—are most strongly associated with landslide susceptibility?

To answer these questions, we construct an improved susceptibility map through a detailed examination of social variables and a new social lag index; that is, we link landslide susceptibility with social variables. Our central objective is to determine whether disadvantaged populations are disproportionately exposed to higher landslide susceptibility. Moreover, to improve the realism of our susceptibility map, we incorporate hydrological terrain datasets derived from digital elevation models (Tau DEM)—approaches partially used in other contexts [17–19]—but not systematically integrated into landslide assessment literature. This is an important addition given that, as global warming intensifies, extreme precipitation events become more frequent, earthflow landslides are expected to occur with heightened frequency [8].

In summary, our objective is twofold. First, we aim to generate a landslide susceptibility map incorporating hydrological and terrain factors, identifying zones of greatest hazard. Second, we quantitatively relate these landslide susceptibility levels to social disadvantage. By focusing on the interconnection between landslide susceptibility levels and social lag indicators, this study addresses a gap in knowledge that has scarcely been examined. Together, these contributions offer new insights into the socio-environmental dynamics of landslide risk and support the development of more integrated disaster risk reduction strategies.

## Literature background

Landslides, also referred to as mass movements or mass wasting, are natural processes that involve the downward movement of soil and rock along slopes. Such processes depend on geological factors and are triggered by events such as rainfall, earthquakes, volcanic eruptions, and weathering [20]. When landslides occur near human settlements, they can threaten livelihoods and economic activities [6]. Landslides are classified according to several criteria, including the material involved and the type and speed of movement. For instance, a fall landslide occurs when a mass detaches from a cliff and moves downward as rock or soil. A flow landslide involves water-saturated ground that moves downhill, typically following intense rainfall [20]. In this study, we use the term "landslide" generically to refer to any type of mass movement.

The Himalayan region is known for having some of the highest peaks on the planet, which in turn has been affected by various landslides due to its orography. In this regard, [6] focus on landslide zonation and the evaluation of social vulnerability in a Himalayan region. To this end, they consider socio-economic variables such as housing, population, illiteracy, and income. They construct a vulnerability index that allows them to calculate landslide risk as a combination of susceptibility and social variables.

These authors conducted their study using a variant of the Frequency Ratio method, called the Analytical Hierarchy Process. They incorporate explanatory variables such as distance to roads, distance to tectonic plates, and Normalized Difference Vegetation Index (NDVI), among others. They found that areas with high susceptibility cover only 13% of the geographical area but concentrate 48% of the landslide occurrences. They construct an index with social variables to determine risk zones, but they do not calculate a statistical association between landslide occurrence and the aforementioned social variables.

Furthermore, [8] integrate socioeconomic and ecological variables to create a vulnerability map for the state of Guerrero in Mexico, which is a mountainous area. Separately, they created a susceptibility map with AutoML, which they

aggregate to establish enhanced-information risk areas; that is, the risk is composed of landslide-prone area indicators in addition to social characteristics. They found high-risk zones located close to low-risk areas. Furthermore, they also found that ecologically vulnerable areas are prevalent in mountainous zones, although high and very-high vulnerability values are dispersed along wide areas of the territory. In addition, socially vulnerable areas are prone to be located in urban and semi-urbanized areas. An important finding is that landslide-susceptible areas do not necessarily overlap with highly vulnerable zones, although they do not present statistical methods to support this association.

Moreover, [7] created a Social Vulnerability Index (SoVI), which they applied to the middle Ribeira region in Brazil, where they used 30 social variables. By using principal component analysis, they built a social vulnerability index and found that the most vulnerable zones to landslides are also the most socially vulnerable, with rural areas being the ones facing higher peril, and social vulnerability is unevenly distributed among municipalities. Furthermore, the authors did not use a statistical method to test that association. In fact, their susceptibility map with SoVI overlays shows that zones with higher susceptibility have fewer inhabitants than otherwise.

Within the context of SoVI works, [21] incorporated 38 variables to build their index and used principal component analysis to get explanatory factors, they found that 36 parishes out of 149 have high to very high social vulnerability. More importantly, the authors found that higher landslide risk is associated with worse social conditions, given that parishes with very high social vulnerability are disproportionately located in high-risk zones.

A related work [22] shows which types of communities are more resilient to landslide occurrence, utilizing social, economic, physical, and institutional variables. They found lower resilience in rural areas as compared to urban zones, with the inner city fringe showing higher values of economic, social, and physical resilience. The second strongest dimension is the urban fringe, and the most fragile is the rural fringe.

Additionally, [23] uses social, ecological, and technological variables as landslide drivers for their landslide assessment in the metropolitan area in Sao Paulo, Brazil. They found that the majority of landslides occurring in cities happen in less developed areas, which turn out to be the most socially vulnerable areas because of the existence of precarious outposts; they estimate 4.6 times higher chances of landslide occurrence in subnormal settlements than in normal ones. Furthermore, they performed a logistic regression model to identify contributing social, physical, and ecological factors.

Regarding the potential relationship between landslide susceptibility and social variables, some studies [6,7,21,23] have found that higher landslide susceptibility correlates with poorer socio-economic conditions, whereas others [8] report the opposite trend, which provides valuable context for our work. In summary, Table 1 presents the most important characteristics of these studies. These works provide a general framework within which our chosen techniques fits.

## Study area

The study area is located in the western outskirts of Mexico City and is particularly relevant because it has undergone irregular urbanization, with stark social disparities: some of the most affluent neighborhoods lie directly adjacent to very poor ones [16]. The area is also characterized by steep hillsides where, due to the social disparities, both wealthy and low-income families have built their homes along steep slopes. However, differences in construction quality, access to social services such as education and healthcare, and basic housing infrastructure could imply that wealthy households face different hazard profiles than their less affluent counterparts who live side by side [13,14].

Specifically, the study region is enclosed within a bounding box between coordinates $(x_{min}, y_{min}) = (-99.40626, 19.30073)$ and $(x_{max}, y_{max}) = (-99.152846, 19.542305)$, with a width of 26,647 meters, a length of 26,688 meters, and a total area of 711,171,175$m^2$. This region comprises a section of Mexico City and a section of the adjacent State of Mexico. Elevations range from a minimum of 2,200 meters above sea level to a maximum of 3,251 meters [24], indicating a considerable altitudinal gradient.

The climate in the area is predominantly temperate, with humid semi-cold and subhumid temperate variants [25]. Precipitation, the main trigger of landslides, exhibits a similar annual pattern in both Mexico City and the State of

**Table 1**. Comparative perspective of studies integrating landslide susceptibility and social vulnerability.

| Study / Region | Methodology | Main Findings |
| --- | --- | --- |
| [6] Himalayas | Analytical Hierarchy Process (AHP), Frequency Ratio | 13% of the land area highly susceptible, containing 48% of landslides; vulnerability index with social variables (housing, income, illiteracy). |
| [8] Guerrero, Mexico | AutoML susceptibility + ecological/social overlays | Identifies mismatches: socially vulnerable zones not always overlapping with susceptibility; ecological vulnerability concentrated in mountains. |
| [7] Brazil (Middle Ribeira Valley) | SoVI using PCA (30 variables) | Rural areas most socially vulnerable and most exposed; uneven distribution of vulnerability among municipalities. |
| [21] Ecuador (149 parishes) | SoVI with 38 variables + PCA | 36 parishes with high–very high vulnerability; high vulnerability parishes disproportionately in high-risk zones. |
| [22] Brazil (resilience) | Composite resilience index (social, economic, physical, institutional) | Inner urban fringe more resilient; rural fringe least resilient. |
| [23] São Paulo, Brazil | Logistic regression + socio-ecological variables | Landslide risk 4.6× higher in informal settlements; strong overlap of low development and hazard. |

**Table notes:** AHP = Analytic Hierarchy Process; AutoML = automated machine learning; PCA = principal component analysis; SoVI = Social Vulnerability Index. Percentages and effect sizes are reported as stated in the original studies e.g., "13% land area" refers to the share of mapped area classified as highly susceptible. Comparisons across studies should be interpreted cautiously because inventories, spatial resolution/extent, variable sets, and index construction differ across regions.

Mexico. In 2023, the most recent year with complete records by the time this work was carried out, only January and February were months without rainfall in both jurisdictions. In the State of Mexico, precipitation increases from March onward, reaching a maximum of 190.2 mm in July and 169.3 mm in August. For Mexico City, total precipitation reached 90.9 mm in July and 57.8 mm in August; these two months concentrated most of the annual rainfall [26] and therefore, correspond to the period with the highest potential for earthflow landslide occurrence.

It is also of capital importance to describe the types of territorial divisions used in this study. In Mexico, the largest administrative territorial division is the state, and our study region includes the State of Mexico. Equally important is Mexico City, which is not itself a state but a federal entity that borders the State of Mexico to the west; functionally, it is equivalent to a state at the federal level.

Below the state level, the next, more granular territorial subdivision is the municipality. The State of Mexico has 125 municipalities, of which Huixquilucan de Degollado and Naucalpan de Juárez are of particular interest in this study. In Mexico City, the equivalent administrative unit to the municipality is the borough (Alcaldía). The city has 16 boroughs, of which Cuajimalpa de Morelos and Álvaro Obregón are relevant for our analysis.

Finally, the most granular level of territorial division we worked with is the Basic Geo-Statistical Area or AGEB. Its boundaries are drawn using physical characteristics such as rivers, lakes, or hills, as well as cultural traits. There are two types of AGEB, rural and urban [27]; in this work we analyzed up to 980 AGEBs.

## Risk atlas

In Mexico, the content of a risk atlas is defined by the General Civil Protection Law, which in Article 112 establishes that it must include a geographic information system, hazard maps, and susceptibility maps, among others [28]. However, despite being a minimum standard for each municipality in terms of the geographic items each atlas must include, in social terms, this minimum is not specified. This results in certain social variables being present in some studies and absent in others.

To begin with, in Mexico City, the Álvaro Obregón borough has 13 officially recognized ravines [29], which are identified as the main locations where landslides may occur, although they are not the only areas potentially affected. The borough recognizes landslides and floods as the greatest hazards to its territory and acknowledges having achieved

only 12% progress in planning for recovery in the event of a landslide [30]. In the borough of Cuajimalpa, 10 ravines are officially recognized [29]. However, unlike Álvaro Obregón, landslide risk is not explicitly recognized; the document only recommends increased surveillance in high areas when precipitation is intense. The word "landslide" or the term "unstable slope" does not even appear in the entire document [31], suggesting the need for improved risk assessment in this borough.

Regarding the State of Mexico, the main landslide trigger in the municipality of Huixquilucan is precipitation [32]. At the time of the atlas's creation, 5 landslides were reported. Additionally, 30 houses are located on steep slopes and ravines where more than 150 people live [33]. In the municipality of Naucalpan de Juárez, 35 AGEBs were identified with a high landslide risk, characterized by steep slopes and bare soil where 171,747 people live; 101 AGEBs were also found with medium risk and a population of 460,377 people [32].

## Social characteristics

A critical issue in this work is the potential for the most vulnerable population groups to be the most affected by landslides. In this sense, it is important to emphasize the inequality observed within each demarcation. In fact, in terms of the Gini coefficient, which measures income distribution inequality, for the year 2018, Mexico City ranks the highest among all federal entities in Mexico with a value of 0.53, and the State of Mexico has a value of 0.4014 [34], within a range of 0 to 1, where a value of 1 implies maximum inequality.

In 2018, one third of the population in Mexico City lived in poverty, while another third was considered vulnerable or at risk of falling into poverty due to lack of access to social services or low income; moreover, households in the highest income decile received 20 times the income of the bottom 20% [35]. For the State of Mexico, 42.7% of the population is living in poverty. Given these levels of inequality, social lag indicators do not necessarily capture the degree of inequality within each borough and municipality; nonetheless, this is the only standardized information available.

In this context, CONEVAL's social lag index is constructed from indicators of social rights, including educational lag, access to health services, housing quality and space, basic housing services, and household goods. The indicator is subsequently categorized into five levels, which are: very low, low, medium, high, and very high [36]. At the AGEB level of granularity considered here, the index is only available in categorical form. Therefore, we used the same underlying variables to construct our own numerical Social Lag Index (SLI) to make it more viable to find numerical associations. In what follows, we describe the social characteristics of each territorial demarcation.

The Álvaro Obregón borough has a total population of 759,137 inhabitants; the majority is young and economically active, between 20 and 29 years of age. The borough is classified as having very low marginalization, however, as noted above, this indicator may not be representative given the internal heterogeneity of the borough, and a more granular analysis is required. In Álvaro Obregón, 64.7% of the population is economically active; 1,153 homes have dirt floors, 417 homes lack piped water, and more than nine people live in 5,114 homes, which are therefore considered highly overcrowded; there are 51 dependents per 100 inhabitants over 65 years old or under 15 years old [30].

In the Cuajimalpa borough, there are 217,686 inhabitants; the median age ranges between 20 and 29 years, meaning that the population is young and economically active. There are 48 dependents per 100 inhabitants; 1,475 homes are occupied by more than nine people; 591 homes have dirt floors; 285 homes lack potable water; and 125 homes do not have drainage. Finally, the social development index of this borough is classified as very low [31], although, as mentioned earlier, due to conditions of inequality, this indicator at the municipal level may not be very reliable, and a higher level of granularity may be necessary.

Concerning the State of Mexico, Huixquilucan municipality has a population of 267,858 inhabitants with a population density of 1,908.7 inhabitants per square kilometer. Of this total, 10% lives in rural communities. The average population growth rate has decreased from 3.93 in the year 2000 to a projected 1 for 2018. In terms of migration, Huixquilucan has been a population attractor in the last 35 years, a consequence of territorial availability, lower land prices compared

 

to Mexico City, and proximity to it, which has caused an increase in housing construction, particularly luxury housing. The migratory balance, understood as the difference between entries and exits, was 182,961, confirming the municipality's role as a population attractor. In terms of access to health services, 23% of the population does not have access to them [33]

The last large municipality within our study area is Naucalpan de Juárez. This is a municipality with a substantially larger population than the demarcations mentioned above, with a total of 833,779 inhabitants. Until shortly before the year 2000, this municipality showed substantial population growth, partly due to migration driven by its proximity to Mexico City; around 10% of the population is of foreign origin [32]. This trend was exacerbated by the limited availability of new residential real estate developments in Mexico City at that time. Likewise, Naucalpan is an industrial area where factories and corporations are located, which created job offers and promoted migration to the municipality.

Regarding marginalization, Naucalpan de Juárez has a general rating category of "Very Low". However, as mentioned earlier, this is due to the phenomenon of inequality, which causes the index to be biased downward, when in reality there are no fewer than 12 localities with high or very high levels of marginalization. Likewise, 32% of the population is at some level of poverty, and 4.3% is in extreme poverty [32].

In sum, the Naucalpan and Huixquilucan municipalities, which are adjacent to Mexico City, have been population attractors both due to the job opportunities offered inside these demarcations and those that their inhabitants can access given their proximity to Mexico City. This, however, has the disadvantage that people migrating to these areas could end up inhabiting zones with elevated landslide risk.

## Materials and methods

**Ethics Statement:** *In this study, no humans or animals were involved or harmed. Furthermore, we did not recruit any human participants, we did not analyze any specific subjects, nor did we analyze animal, medical, or biological samples. We did not conduct prospective or retrospective studies of any kind and we did not apply any questionnaire; we did not use medical records or archived samples. We did not use any animal models and did not perform tests on living subjects. We exclusively used public datasets that do not identify any individual and, consequently, it is not possible to harm individuals by exposing personal information, as we did not obtain any personal data.*

In this section, we present the methods used to perform the calculations. Specifically, the methods presented here were used to carry out the following tasks:

- Create the landslide Susceptibility Map.
- Create the Social Lag Index (SLI).
- Find a potential association between the landslide susceptibility and SLI.

### Models used for the susceptibility map creation

To identify the most appropriate method for producing the susceptibility map, we tested 13 models, ranging from simple Weight-of-Evidence approaches to advanced deep neural network–based models (Deep Learning). A brief description of each model is presented below:

**Extreme gradient boosted trees (XGBoost).** XGBoost is a highly efficient implementation of gradient-boosted decision trees. It builds additive models which are nonparametric regression models that approximate the response surface as a sum of smooth functions by combining weak learners and iteratively correcting the errors of previous trees. It is useful for capturing nonlinear relationships between landslide conditioning factors such as aspect, and land cover. XGBoost parallelizes tree boosting, making it computationally efficient [37]. Its complexity depends on the number of trees, number of features, tree depth, and data size. Its parallel architecture translates even complex calculations into relatively simple computational threads. Although efficient, XGBoost has several limitations: it includes hyperparameters

that must be tuned, can be computationally intensive, and may overfit the data even with regularization. Finally, it is often regarded as a black-box model since the modeler is not fully aware of its internal calculations [37].

Note: We anticipate that this was the champion model in the upcoming application. Therefore, we present the model hyperparameters that were used:

- Number of estimators: 100
- Maximum depth: 6
- Learning rate: 0.1
- Ratio of number of negative class to the positive class for unbalanced classes (scale pos weight):

$$\frac{|ytrain| - \Sigma ytrain}{\Sigma ytrain}$$

- No label encoder used
- Evaluation metric: logloss

**Light gradient boosted machine (LightGBM).** LightGBM is similar to XGBoost in that it is a highly optimized tree-based boosting method. It uses a histogram-based learning algorithm and grows trees leaf-wise, which can increase accuracy and efficiency [38].

**CatBoost.** CatBoost is a gradient boosting algorithm that uses an ordered boosting scheme and permutation strategies to reduce variance and overfitting. It is well suited to handling categorical data, which is useful for some landslide conditioning factors [39].

**Random forest.** Random Forest builds an ensemble of decision trees and averages their outputs. In this sense, the method is a typical ensemble learner. This aggregation process reduces variance and tends to mitigate overfitting [40].

**Deep learning.** Deep learning refers to the use of neural networks with more than two hidden layers that automatically learns hierarchical features. It is well suited to modeling nonlinear relationships, although it is computationally intensive. In landslide studies, it often involves the combined use of convolutional neural networks for feature extraction followed by deep neural networks for classification [41].

**Generalized additive model (GAM).** A Generalized Additive Model (GAM) is a flexible regression framework in which the response is modeled as a sum of smooth functions. This approach provides a robust way to capture nonlinear relationships. GAMs are well suited to representing the influence of topographic and climatic variables without imposing predefined functional forms [42].

**K-nearest neighbors (KNN).** The K-Nearest Neighbors (KNN) algorithm is an instance-based method in which predictions are made by voting or averaging the closest $k$ data points. Consequently, the algorithm classifies susceptibility levels according to the similarity of neighboring zones. It is a nonparametric method and can be sensitive to imbalanced data, making careful data pre-processing essential [43].

**Frequency ratio.** The Frequency Ratio (FR) method is a statistical technique that is simple to implement and often yields accurate results. The frequency ratio is defined as the ratio of the area where landslides have occurred to the entire study area, that is, the ratio of the probability of landslide occurrence to the probability of non-occurrence for a given class of an explanatory variable. The susceptibility map is obtained by summing the contributions of all attributes [44].

**Weight-of-evidence.** The Weight-of-Evidence method is a Bayesian approach that assigns weights to predictor classes based on their likelihood ratios. In this sense, each variable adds or subtracts weight from the probability of landslide occurrence at a specific location [45].

**Naive Bayes.** Naive Bayes computes the probability of landslide occurrence using Bayes' theorem under the assumption of conditional independence among predictors. The model is termed "naive" because this assumption is often violated in practice [46].

**Support vector machine.** A Support Vector Machine (SVM) is a margin-based classifier that finds the optimal hyperplane that maximizes the margin between classes. By using kernel functions, SVM can project data into higher-dimensional spaces to find nonlinear boundaries between landslide-prone and safe zones. In essence, it draws boundaries between landslide-prone and relatively safe zones using support vectors, which define the separating margin. The method seeks the widest possible separation between these support vectors [47].

**Maximum entropy.** The Maximum Entropy (MaxEnt) model estimates the most uniform distribution of class probabilities subject to specified constraints. It performs well with limited data, as it constructs the least biased model consistent with the information provided by the model inputs or conditioning factors [48].

**Logistic regression.** Logistic regression models the log-odds of landslide occurrence as a linear combination of causative factors. The method is parametric, as it assumes that the underlying relationship can be described by a finite number of parameters [49].

Given that we evaluated a considerable number of models, it was not feasible to provide an exhaustive description of each. Instead, Table 2 summarizes their most relevant characteristics.

## Performance metrics

Once each of the aforementioned models was fitted, the next step was to compare their performance. To this end, we used the Receiver Operating Characteristic (ROC) curve and the Area Under the Curve (AUC). The ROC curve provides a visual representation of classifier performance across all possible thresholds. It plots the true positive rate on the y-axis and the false positive rate on the x-axis [50].

The ROC curve compares the true positives with the false positives. A point on the curve is preferable to another if it lies closer to the upper-left corner, indicating a low false positive rate and a high true positive rate; therefore, a classifier whose ROC curve lies above the diagonal identity line ($y = x$) is better than random guessing. To make ROC-based comparisons more concise, we summarize performance using the area under the ROC curve (AUC). AUC takes values between 0 and 1, and no useful classifier should have a value below 0.5, which corresponds to random performance [50].

## Social lag index construction

As mentioned earlier, CONEVAL's social lag index is only available as categories at the AGEB level of granularity. Consequently, to obtain a more accurate measure of association between landslide susceptibility and the SLI, we created our

**Table 2**. General characteristics of landslide susceptibility models.

| Model | Type | Norm. | Non-lin. | Advantages |
|---|---|---|---|---|
| XGBoost | Boosted trees with gradient optimization | No | ✓ | Handles missing data, feature importance, robust to multicollinearity |
| LightGBM | Leaf-wise boosted trees with histogram bins | No | ✓ | Fast on large datasets, supports categorical data |
| CatBoost | Ordered symmetric boosting for categoricals | No | ✓ | Minimal preprocessing, less tuning |
| Random Forest | Bagging ensemble of decorrelated trees | No | ✓ | Reduces overfitting, variable importance |
| Deep Learning | Multi-layered neural networks (DNN/CNN) | No | ✓ | Captures hierarchical patterns, suited for image/terrain |
| GAM | Additive model with smooth terms | No | ✓ | Interpretable, handles spatial/non-linear patterns |
| KNN | Instance-based learner by proximity | No | ✓ | Adapts to local structure, no training |
| Frequency Ratio | Heuristic ratio of occurrences | No | ✓* | Simple, reproducible, interpretable |
| Weight-of-Evidence | Bayesian log-likelihood | No | ✓ | Interpretable, good for categoricals |
| Naive Bayes | Probabilistic, assumes independence | Yes | ✓* | Works with small data, fast |
| SVM | Margin-based classifier with kernels | No | ✓ | Effective in high dimensions, overfitting-resistant |
| MaxEnt | Probabilistic, maximizes entropy | No | ✓ | Suited for rare-event modeling |
| Logistic Regression | Linear log-odds model | Yes | ✓* | Interpretable coefficients, robust baseline |

**Table notes:** "✓*" indicates limited or discrete nonlinearity. All models support nonlinear behavior to varying extents, except logistic regression and Naive Bayes, which assume linearity or independence unless transformed.

own continuous numerical social lag index. To make our index comparable, we used exactly the same variables as those used by CONEVAL, which are the following:

- Total population,
- Occupied homes,
- Low education among people aging 15 years or older,
- Population aged 15-24 not attending school,
- Population without Health Coverage,
- Overcrowded Housing,
- Houses Without a Toilet,
- Houses Without a Washing Machine,
- Houses Without a Refrigerator,
- Houses Without a Landline,
- Illiterate population aged 15 or older,
- Out of School Population aged 6-14,
- Houses With Dirt Floor,
- Houses Without Piped Water,
- Houses Without Drainage,
- Houses Without Electricity,

To begin with, a data normalization was carried out: let $\mathbf{X} = \{x_{ij}\} \in \mathbb{R}^{n \times p}$ be the original feature $n \times p$ matrix with real coefficients, where $n$ represents the number of AGEBs, and $p$ represents the number of social lag–associated variables. Each of the aforementioned variables is normalized using a Min-Max scaler to map values into the interval [0,1] using the following formula:

$$x_{ij}^{norm} = \frac{x_{ij} - \min(x_j)}{\max(x_j) - \min(x_j)} \quad \forall i \in \{1, ..., n\}, j \in \{1, ..., p\}$$

After re-scaling each column, a feature importance analysis was carried out via random forest regression. This model approximates the underlying mapping to predict a target $y_i$ using the variables that were rescaled.

$$y_i = f(x_{i1}, x_{i2}, ..., x_{ip}) + \varepsilon_i$$

The importance of each feature $j$ is quantified by the Random Forest via its average contribution to impurity reduction across all trees, denoted $\phi_j$. These importances are then normalized to form weights:

$$w_j = \frac{\phi_j}{\sum_{k=1}^{p} \phi_k}$$

with the constraint $\sum_{j=1}^{p} w_j = 1$ and $w_j \geq 0$. Once the feature importance weights are known, we need to weight each variable according to these values. Let $\mathbf{w} = (w_1, ..., w_p)^\top$ be the normalized weight vector, and let $\mathbf{x}_i^{norm}$ be the $i$-th row of the normalized data matrix. The Social Lag Index for each spatial unit $i$ is the following weighted sum:

$$SLI_i = \sum_{j=1}^{p} w_j \cdot x_{ij}^{norm} = \mathbf{x}_i^{norm} \cdot \mathbf{w}$$

We emphasize that this is a numerical index, which contrasts with the original CONEVAL categorical index. Our newly built index, in principle, could allow us to model more accurate association relationships between landslide susceptibility and the SLI. This is because, when dealing with a categorical index, if its classes are not equally or nearly equally balanced, one category alone may result very important while the others are not, which usually ends up in a statistically weak regression.

## Association methods

Once the susceptibility maps were created and evaluated, and the social lag index was built, the next step is to find whether there exist a relationship between the social lag index and landslide susceptibility. In this section, we present a method called changepoints detection which is used to partition the dataset in a way that allows us to discern the aforementioned relationship. Additionally, we present two methods to establish a relationship between landslide susceptibility and social lag: SHAP values as an exploratory method, and Copulas as a robust method to detect non-linear relationships between landslide susceptibility and social lag.

**Changepoint detection.** Since the study area is highly heterogeneous, we partitioned the complete zone using two complementary approaches. First, we used a decile-based subdivision; and second, we employed a changepoint detection algorithm to identify abrupt shifts in the underlying process. Although the changepoint algorithm is not itself an association method, its implementation was necessary to enable a more accurate assessment of potential associations.

Changepoints are locations within a time series or ordered sequence where the statistical properties, such as mean, variance, or distribution, undergo abrupt modifications. Their estimation relies on optimizing a cost function that balances model fit and complexity. We used the PELT Algorithm, which stands for Pruned Exact Linear Time. It finds the optimal number of partitions by minimizing a penalized cost function, which balances how good each segment fits. The pruned aspect of the method refers to the fact that it is computationally optimized for efficiency [51].

High-level functionality:

- A cost function is defined for a segment of data.
- The method minimizes the sum of costs for all segments, plus a penalty proportional to the number of changepoints to prevent overfitting.
- At each step, it considers all possible previous changepoint locations and calculates the cumulative cost using dynamic programming.

**SHAP values and Beeswarm plots.** As an exploratory method that allowed us to infer a possible association direction and strength, we used SHAP values. SHAP values, which is short for for SHapley Additive exPlanation values, provide a theoretically grounded framework to interpret complex machine learning models by attributing to each predictor its marginal contribution to a prediction. In essence, SHAP values quantify the direction and magnitude of association between a feature and the model output, while preserving additivity and consistency. A beeswarm plot, which arranges SHAP values across all observations, offers an immediate visualization of feature importance and effect heterogeneity. In such plots, the horizontal spread indicates the strength of association, while color gradients depict the underlying feature values, thereby enabling a first inspection of non-linear associations and potential thresholds [52].

**Copulas.** Once a first exploratory visualization was made using SHAP plots, a confirmatory robust method is needed to detect association between variables, which is the role of copula modeling. A Copula is a multivariate probabilistic construction which, among other characteristics, is capable of binding several different univariate distribution functions into a single multivariate distribution function, regardless of its initial marginal distribution [53].

This is an essential characteristic, since by using copulas it is possible to find relationships between apparently heterogeneous variables. Moreover, given that this method is capable of modeling each variable as it is and then joining them, it is an essential tool to preserve data's natural structure and relationships, that is, without applying any transformation to

force data into normality, for example. Because of these Copula characteristics, they are an invaluable instrument when nonlinear, extreme, and asymmetric dependence structures arise.

According to Sklar's Theorem any multivariate joint distribution $H(x_1, \dots, x_d)$ with continuous marginals $F_1(x_1), \dots, F_d(x_d)$ can be uniquely decomposed as:

$$H(x_1, \dots, x_d) = C(F_1(x_1), \dots, F_d(x_d)),$$

where $C : [0,1]^d \to [0,1]$ is the copula. This decomposition separates the dependence structure captured by the copula from the marginal distributions, which allows for much flexibility when modeling intricate relationships.

Copulas are particularly valuable in fields where nonlinear and tail dependencies are critical, such as finance, hydrology, and macroeconomics. They allow researchers to model joint distributions even when marginals differ, capture asymmetric dependencies, and quantify risks associated with extreme co-movements. There are several copula families that aim to model different behaviors. Some of the best-known representations are the following:

- **Elliptical copulas:**
  - Gaussian copula: symmetric and not necessarily with heavy tails.
  - Student-t copula: similar to Gaussian but includes symmetric tail dependence.
- **Archimedean copulas:**
  - Clayton copula: captures strong lower tail dependence (joint low events).
  - Gumbel copula: captures strong upper tail dependence (joint upper events).
  - Frank copula: symmetric, flexible for moderate dependence.

To summarize, some of the benefits of using copulas are that they are flexible since arbitrary marginals can be combined with a wide variety of dependence structures. Also, they allow researchers to model tail dependence, and they capture joint extremes that correlation alone cannot. Furthermore, complex dependence is summarized by a few parameters (e.g., copula parameter theta for dependence, Kendall's$\tau$, tail indices). Table 3 summarizes copula families that will be used in this work.

**Computational specifications.** The complete modeling was carried out in Python 3.11 and 3.12. We ran the models in two settings. One on a local implementation using a MacBook M2 with 8 cores and 16 core Neural Engine, in which we estimated the Machine learning models. The computation time to run the whole setting, including every model as well as its testing variants was about an hour. The second setting is a Google Colab implementation with normal runtime i.e. we didn't use GPUs provided the uptime is shorter. Here we did data pre-processing as well as the complete association computations: SHAP, Copulas, Changepoints, Deciles, etc. It took roughly an hour and a half to perform every needed calculation.

**Table 3**. **Common copula families and their bivariate properties.**

| Copula | Param. | Tail Dep. | Distribution Form C(u,v)= |
|---|---|---|---|
| Gaussian (Normal) | $\rho \in (-1, 1)$ | None | $\Phi_\rho(\Phi^{-1}(u), \Phi^{-1}(v))$ |
| Student-t | $\rho, \nu > 2$ (df) | Symmetric | $t_{\rho,\nu}(t_\nu^{-1}(u), t_\nu^{-1}(v))$ |
| Clayton | $\theta > 0$ | Lower tail | $\left(u^{-\theta} + v^{-\theta} - 1\right)^{-1/\theta}$ |
| Gumbel | $\theta \geq 1$ | Upper tail | $\exp\left[-\left((-\ln u)^\theta + (-\ln v)^\theta\right)^{1/\theta}\right]$ |
| Frank | $\theta \neq 0$ | None | $-\frac{1}{\theta}\ln\left(1 + \frac{(e^{-\theta u}-1)(e^{-\theta v}-1)}{e^{-\theta}-1}\right)$ |

**Table notes:** $\Phi$ denotes the standard normal CDF and $\Phi^{-1}$ its quantile function; $\Phi_\rho$ is the bivariate normal CDF with correlation $\rho$. Also, $t_\nu$ and $t_\nu^{-1}$ are the univariate Student-$t$ CDF and quantile, and $t_{\rho,\nu}$ is the bivariate Student-$t$ CDF with correlation $\rho$ and degrees of freedom $\nu$. Parameter limits yielding independence: Clayton $\theta \to 0$, Gumbel $\theta = 1$, Frank $\theta \to 0$. For the Student-$t$, $\nu > 2$ ensures that finite variance.

Furthermore, computational time varies immensely depending on the local implementation hardware used and if the computation is parallelized. Furthermore, Colab calculation also varies a lot depending on the cloud available resources, as well as on the account type: paid accounts have faster runtimes. These facts make any computational time relative and dependent on the factors just mentioned.

## Data sources

All spatial inputs were restricted to the study area defined by the bounding box with coordinates $(x_{min}, y_{min})$ = $(-99.40626, 19.30073)$ and $(x_{max}, y_{max}) = (-99.152846, 19.542305)$. The reference coordinate system adopted for all layers is WGS 84 / UTM zone 14N (EPSG:32614) to ensure spatial consistency across datasets. The final grid resolution was set to 30 m, which approximately matches the native resolution of most environmental inputs.

We used a Digital Elevation Model (DEM) from the United States Geological Survey (USGS) via OpenTopography [54], based on the 2007 release but updated with data acquired in 2017 for the study area. The DEM is distributed at 30 m (1 arc-second) resolution and was provided as an `int16` raster with a NoData value of −19999. From this source, we got slope, aspect, and roughness layers at the same resolution. Additionally, as part of the essential inputs for the model, slope and aspect rasters provided by USGS [54] were incorporated at the same resolution and spatial extent.

Land cover information was obtained from the Global Land Analysis and Discovery (GLAD) program at the University of Maryland [55]. We used the 2000–2020 collection (30 m resolution), which provides maps at five-year intervals. For this study, we used the 2020 layer, reprojected and resampled to match the DEM grid. Bicubic interpolation (`order=3` with anti-aliasing) was applied during resizing to minimize spectral distortion. Although the DEM inputs are dated to 2017, there has been no major disturbance (e.g., severe drought or a cyclone directly impacting the area) that would substantially modify the terrain, so we expect the impact of this temporal offset on the analysis to be negligible.

Landslide inventory was obtained from the World Bank Global Landslide Catalog [56], covering events from 1980 to 2018. Original data (median native resolution ∼1 km) were rasterized, resampled, and reprojected to 30 m to match USGS data. Nearest-neighbor resampling was applied to avoid artificial smoothing of categorical landslide presence. Although this downscaling improves spatial alignment with other layers, we acknowledge that the original 1 km resolution limits the precision of landslide representation.

Furthermore, Mexico's territorial division shapefiles for every AGEB in Mexico City and the State of Mexico were downloaded from Mexico's National Institute of Statistics and Geography (INEGI) [57]. Each shapefile contains multiple AGEB divisions for the State of Mexico and Mexico City and we retained only those intersecting the bounding-box–delimited study area. To associate spatial data with social inequality statistics, data from the National Council for the Evaluation of Social Development Policy (CONEVAL) were compiled from its 2020 edition of the Social Lag Degree site [58].

Invalid no-data values were removed using a common mask, and any remaining missing values in predictor variables were imputed with their mean across valid observations; this approach assumes missingness at random and may overlook spatial correlation. As stated, we used subsampling to reduce computational demands by limiting the dataset to a predefined maximum number of pixels, employing stratified sampling to maintain class balance between landslide and non-landslide instances and thereby mitigating potential bias without introducing additional missing data.

Additional hydrological layers derived with the TauDEM (Terrain Analysis Using Digital Elevation Models) toolbox were also incorporated into the analysis. To underscore the relevance of this tool suite, we describe its main components in the following subsection.

**Hydrological tools.** To improve the quality of the susceptibility map, we incorporated the Terrain Analysis Using Digital Elevation Models (TauDEM) suite from the USGS. Moreover, several of these hydrological inputs have already been used in related landslide studies [17–19]. These tools are particularly useful for enhancing the representation of earthflow landslides, which are expected to occur more frequently in areas with moderate to high precipitation, such as our study region. As noted in the Study Area section, up to 190.2 mm of rainfall can occur during the wettest months, and this amount is likely to increase under climate change [59].

Tau DEM inputs were constructed using the D8 and D-Infinity algorithms. The Deterministic 8-neighbor (D8) algorithm is used to define the flow direction within a digital elevation model by identifying the maximum downslope two-dimensional gradient to calculate the flow direction for each cell. Specifically, as a grid-based method, it evaluates the flow direction from a central cell to one of its eight neighboring cells (north, northeast, east, southeast, south, southwest, west, and northwest) [60]. This algorithm is useful for our work, since we are trying to strengthen the robustness of the susceptibility map to earthflow slides.

Furthermore, the D8 algorithm systematically applies the previously described process across the entire grid, modeling the natural flow of water over the terrain. However, it is worth noting that while the D8 algorithm is computationally efficient, it does have limitations. In flat terrains or regions with minimal elevation changes, its deterministic nature may lead to unrealistic flow patterns by forcing flow to a single neighbor, ignoring the possibility of dispersion.

In addition, the D-Infinity algorithm, offers an advanced approach to flow direction determination in DEMs. Unlike the D8 method, the D-Infinity algorithm allows flow to be proportionally distributed among two neighboring cells rather than being directed to just one [61]. This method calculates the flow direction based on the steepest downslope direction derived from elevation gradients, treating the flow as a continuous field. By accounting for the terrain's aspect, the algorithm determines the flow direction as an angle and distributes the flow proportionally to the two adjacent cells that align most closely with this angle.

Moreover, by distributing the flow between two neighboring cells, the D-Infinity algorithm addresses some of the limitations inherent in the D8 approach, particularly in representing divergent flow over convex terrain and convergent flow in concave areas. This proportional distribution leads to more realistic modeling of hydrological processes, especially in areas with gentle slopes or complex topography. Consequently, the D-Infinity algorithm enhances the accuracy of flow accumulation calculations and watershed delineations, providing a more nuanced representation of how water and materials move across the landscape [61]. The inputs above have the following characteristics, according to [62]:

1. **D8 Contributing Area**: Refers to the contributing area in terms of the number of grid cells. This tool assesses edge contamination, which is the chance that a contributing area may not be as representative as it should be, given that other cells outside its domain may not be taken into consideration.
2. **D8 Flow Direction**: It is a grid that provides the flow direction given by the D8 method.
3. **D8 Slope**: Comprises a grid in the D8 flow direction measured in drop per unit distance.
4. **D-Infinity Contributing Area**: Calculates the contributing area per unit contour length, assuming the flow is evenly distributed.
5. **D-Infinity Flow Direction**: Determines the flow direction using the D-Infinity method, which outputs a grid with direction angles.
6. **D-Infinity Slope**: A slope grid evaluated using the D-Infinity method.

Of the 12 inputs considered, 10 were obtained directly from the United States Geological Survey (USGS) at a standard resolution of 30 m. Consequently, only the remaining two layers (land cover and the landslide mask) required substantial transformation to match USGS specifications, as summarized in Table 4.

Additionally, Fig 1 displays all environmental inputs, including terrain-associated variables such as slope, aspect, and roughness; hydrological variables derived from the TauDEM suite; land cover; and the landslide mask, which together capture topographic, geomorphological, and land-cover characteristics.

The complete data flow and the models incorporated in the analysis are summarized in Fig 2. In this diagram, the leftmost boxes represent the inputs to the landslide susceptibility model, which are passed to the algorithm to generate the susceptibility map. The resulting map is then intersected with the INEGI territorial division and the CONEVAL social-variable database, after which statistical models are applied to extract the most relevant information from this enriched dataset.

**Table 4. Spatial and social input variables used in the landslide susceptibility analysis.**

| Input | Resolution | Transformation / Processing | Source |
|---|---|---|---|
| Elevation (DEM) | 30 m | Reprojected to EPSG:32614; NoData = –19999 | USGS |
| Slope | 30 m | Derived from DEM | USGS |
| Aspect | 30 m | Derived from DEM | USGS |
| Roughness | 30 m | Derived from DEM | USGS |
| D8 Contributing Area | 30 m | Derived from DEM using TauDEM | USGS |
| D8 Flow Direction | 30 m | Derived from DEM using TauDEM | USGS |
| D8 Slope | 30 m | Derived from DEM using TauDEM | USGS |
| D-Infinity Contributing Area | 30 m | Derived from DEM using TauDEM | USGS |
| D-Infinity Flow Direction | 30 m | Derived from DEM using TauDEM | USGS |
| D-Infinity Slope | 30 m | Derived from DEM using TauDEM | USGS |
| Land Cover (2020) | 30 m | Reprojected and resampled with bicubic interpolation | GLAD |
| Landslide Inventory Mask | ~1 km (native) → 30 m | Rasterized, resampled (nearest neighbor), and reprojected | World Bank |
| AGEB Boundaries | Vector (census blocks) | Cropped to bounding box, reprojected | INEGI |

**Table notes:** All raster layers were harmonized to a final resolution of 30 m and reprojected to WGS 84 / UTM Zone 14N (EPSG:32614). Continuous variables were resampled using bilinear or bicubic interpolation, while categorical variables (landslides, land cover) were resampled with nearest-neighbor to avoid spectral distortion.

## Results

By testing 13 different landslide susceptibility models and incorporating hydrological causative factors, we achieved the following:

1. Create a landslide susceptibility map that accounts for the potential paths of earthflow landslides, which is particularly relevant given projected increases in rainfall in some regions due to climate change.
2. Map the AGEB territorial divisions according to their median landslide susceptibility.
3. Associate each AGEB's susceptibility with its corresponding social lag index, as well as with the individual variables used to construct that index.

### Landslide susceptibility maps

Each model used to build the susceptibility map was trained with the 12 aforementioned inputs, including the hydrological variables, with the aim of improving modeling capacity, particularly with respect to earthflow landslides. Fig 3 presents the correlation matrix between these base inputs. The matrix reveals several noteworthy patterns. Land cover contributes largely non-redundant information, while the landslide inventory shows positive correlations with slope, roughness, and elevation, suggesting that steeper and more irregular terrains are more prone to landslides, as expected. At the same time, some predictors display substantial correlations with one another, raising concerns about multicollinearity. To address this, we proceeded under the premise that if the best-performing susceptibility model proved sensitive to multicollinearity, we would reduce the set of inputs to minimize collinear effects; otherwise, if the model demonstrated resilience, the analysis would be carried out using the full set of variables.

To identify which models were most sensitive to multicollinearity, Table 5 reports the AUC metric for each estimated model, together with a qualitative assessment of the multicollinearity effect. The XGBoost model is the best performing model with an AUC of 0.883, which is marginally above the second best performing model, LightGBM, with an AUC of 0.879. In addition, since XGBoost is the champion model, Table 5 also shows a non-significant effect of multicollinear variables, that is, the model is robust to correlated predictors, which is why the study was carried forward.

Furthermore, Fig 4 depicts the ROC curves for every tested model. It is clear that tree-based models (e.g. XGBoost, Light GBM, and CatBoost) are among the best performing methods, since the ROC curves approach the top-left corner, a sign of strong classifier performance.

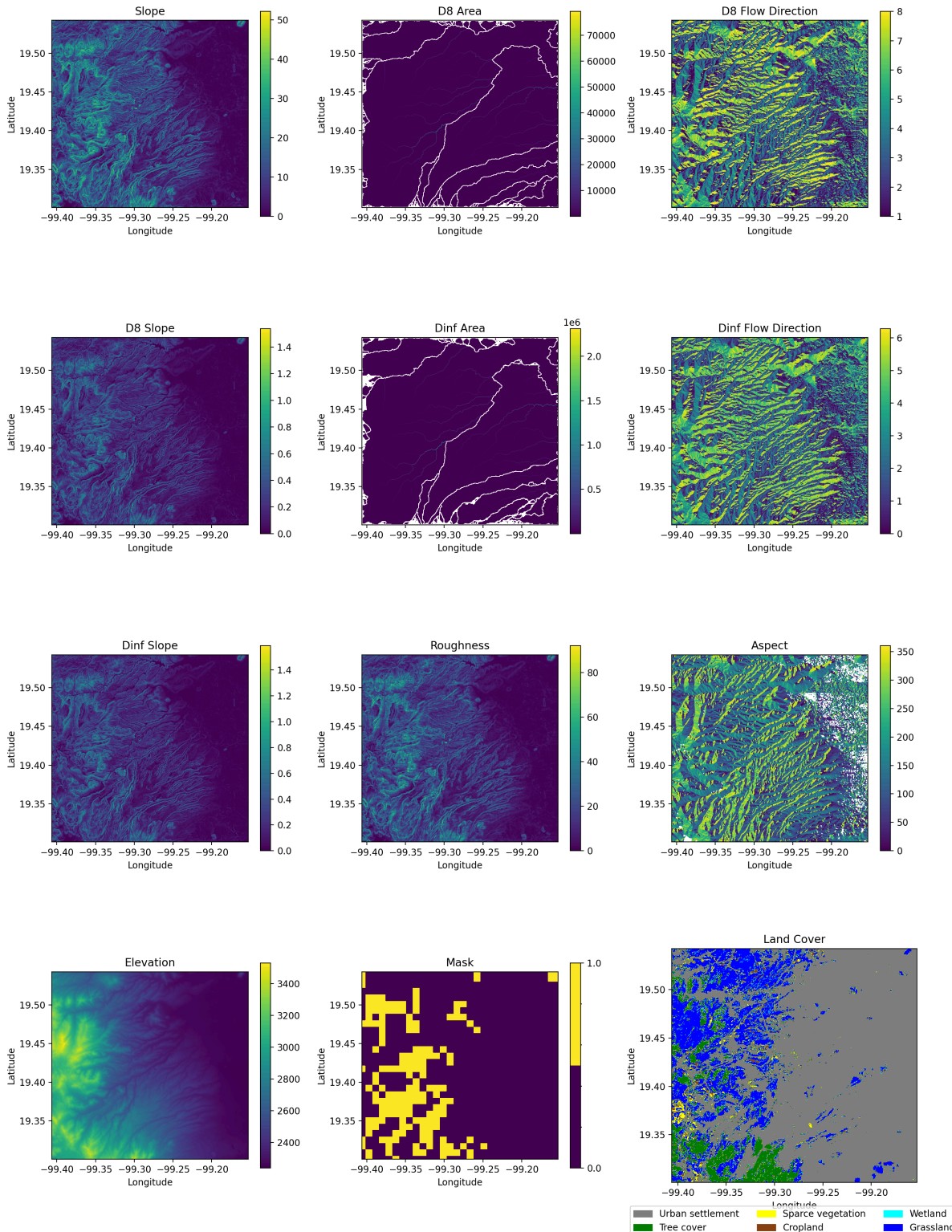

**Fig 1. Inputs used in the estimation of the landslide susceptibility map.** Inputs include DEM, aspect, roughness, land cover, tau DEM and landslide mask.

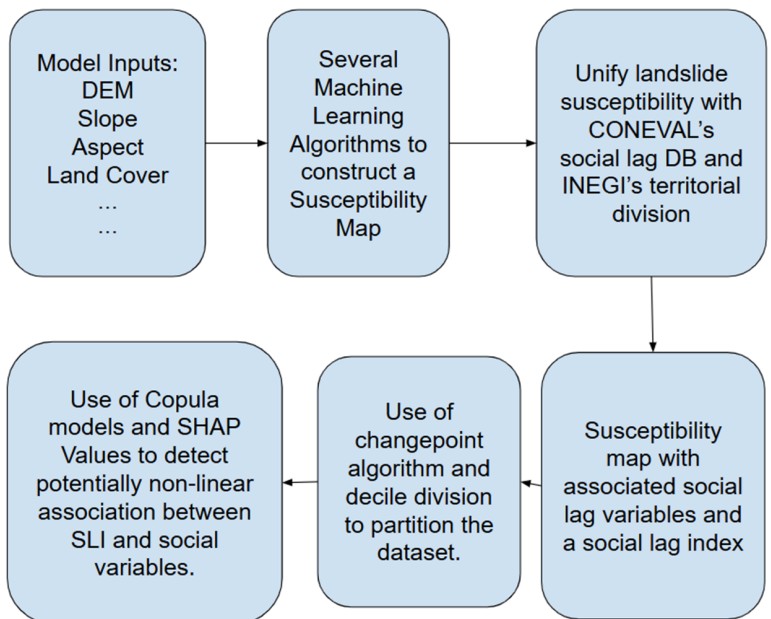

**Fig 2**. **Model information flow.** The diagram show how inputs are transformed into landslide susceptibility maps through several algorithms, evaluated, and subsequently associated with socioeconomic variables.

It is worth mentioning that, despite the fact that a standard procedure to discriminate between models is to use a standard metric, as we did with ROC/AUC, a visual inspection of the output maps is necessary. Output plots shown in Fig 5 depict that there is a high degree of variability among the considered methods. Some of them highlight the top of the mountainous region alone as risky (CatBoost, GAM, KNN, SVM), whereas others consider eastern regions as well. Additionally, regardless of their ROC/AUC value, different maps might serve different purposes. For instance, if it is important to study ravines located along the foothills running east–west, a map that emphasizes this region should be used, such as those constructed with Weight-of-Evidence, Deep Learning, or Maximum Entropy.

Once the best model had been selected, Fig 6 shows the final landslide susceptibility map using the Extreme Gradient Boosted Trees method. Regions with higher landslide occurrence probability are colored in yellow, whereas regions with lower probability are colored in dark blue. This map contains pixel-by-pixel percentile susceptibility, which allows us to obtain, at the map's resolution, the exact susceptibility value for each point in the study area. The highest landslide susceptibility values are located to the west, high in the mountains, while the lowest values are found to the east.

As a way to quantify uncertainty (UQ) in the susceptibility map, we adopted a bootstrap resampling framework, which generates an ensemble of predictions to capture variability arising from sampling and model training processes. Specifically, we performed 50 bootstrap iterations, wherein each iteration involves resampling the subsampled dataset with replacement to form a bootstrapped training set, followed by fitting an XGBoost classifier and predicting susceptibility probabilities across the cleaned feature matrix. The collection of these predictions enables the derivation of ensemble statistics: the maximum and minimum values shown in Fig 7, which represent the upper and lower bounds of prediction variability. These Max and Min maps make it straightforward to show the whole range of variability within the output.

Furthermore, a simplified version of the susceptibility map is its categorical version, that first classifies susceptibility according to ranges. Chosen ranges in this study are shown in Table 6. The purpose of this map is to call attention to specific susceptibility zones by using differentiated or asymmetric ranges. We want to emphasize that the ravine zones on the foothills and the places near the top of the mountain are the most dangerous, as depicted in Fig 8.

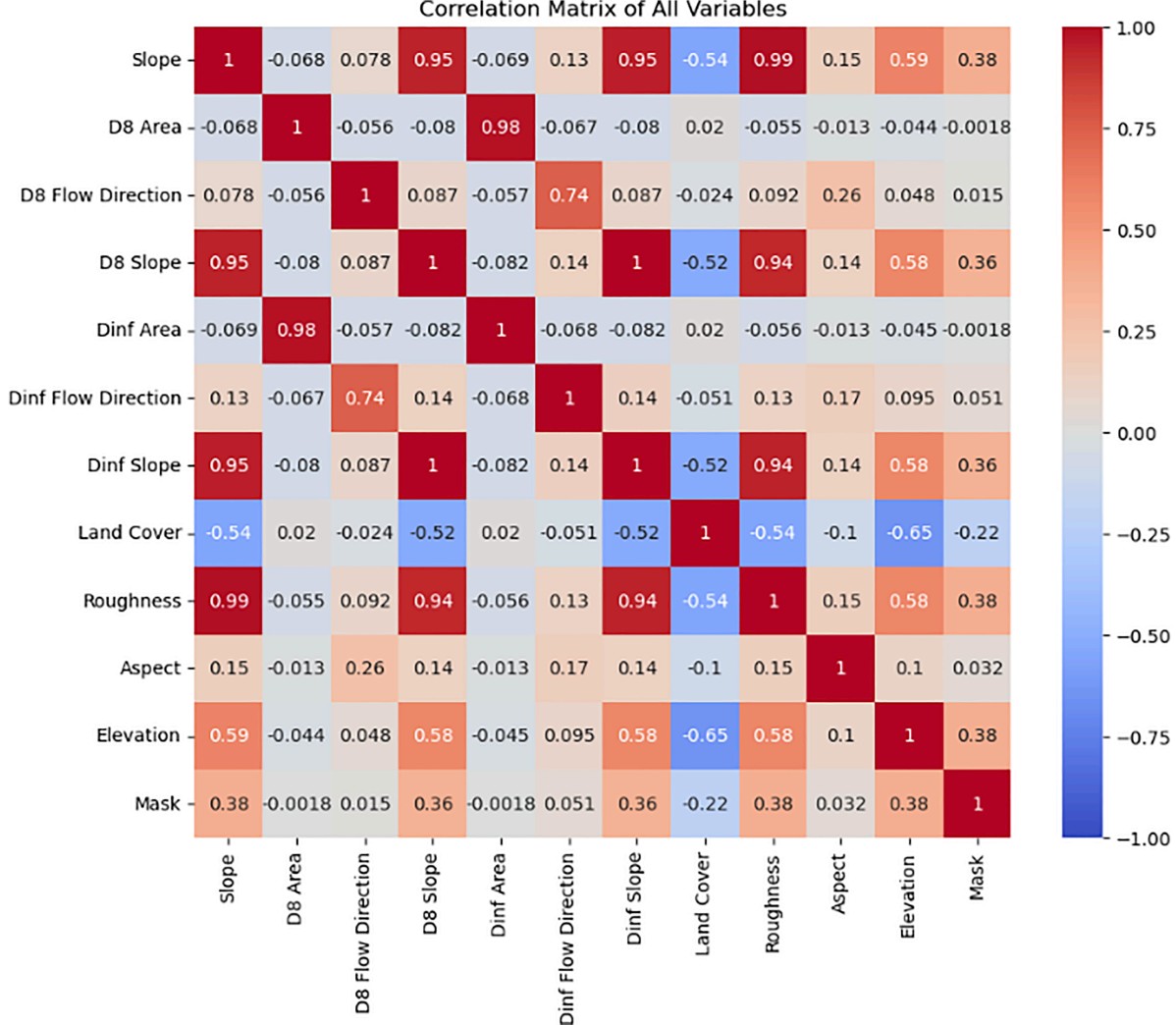

**Fig 3**. Correlation matrix for every causative factor.

**Table 5. Model performance based on AUC values and sensitivity to multicollinearity.**

| Model | AUC | Multicollinearity Effect | Reference |
|---|---|---|---|
| XGBoost | 0.883 | Not significantly affected | [37] [63] |
| LightGBM | 0.879 | Not significantly affected | [38] |
| CatBoost | 0.871 | Affected | [39] |
| Random Forest | 0.870 | Not significantly affected | [40] |
| Deep Learning | 0.870 | Affected | [41] |
| Generalized Additive Model | 0.857 | Affected | [42] |
| K-Nearest Neighbors | 0.816 | Affected | [43] |
| Frequency Ratio | 0.808 | Affected | [45] |
| Weight-of-Evidence | 0.806 | Affected | [46] |
| Naive Bayes | 0.792 | Affected | [47] |
| Support Vector Machine | 0.790 | Mildly affected | [48] |
| Maximum Entropy | 0.758 | Relatively robust | [49] |
| Logistic Regression | 0.710 | Affected | [50] |

**Table notes:** The AUC metric reflects the ranking quality of each model. Multicollinearity primarily affects interpretability and the reliability of models that assume feature independence or linearity.

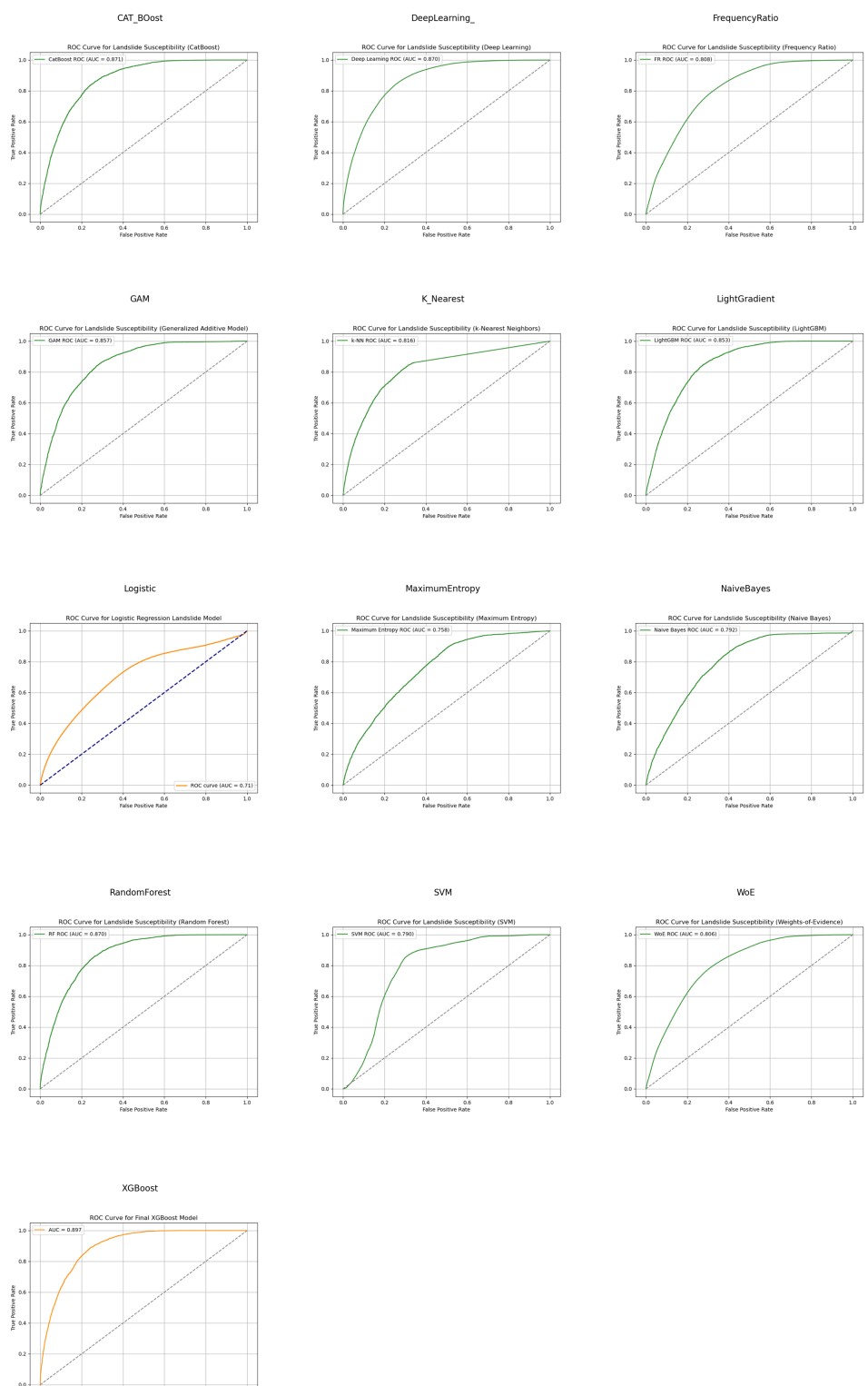

**Fig 4**. **AUC and ROC plots for every tested model.** This matrix plot shows every AUC and ROC for easy comparison purposes.

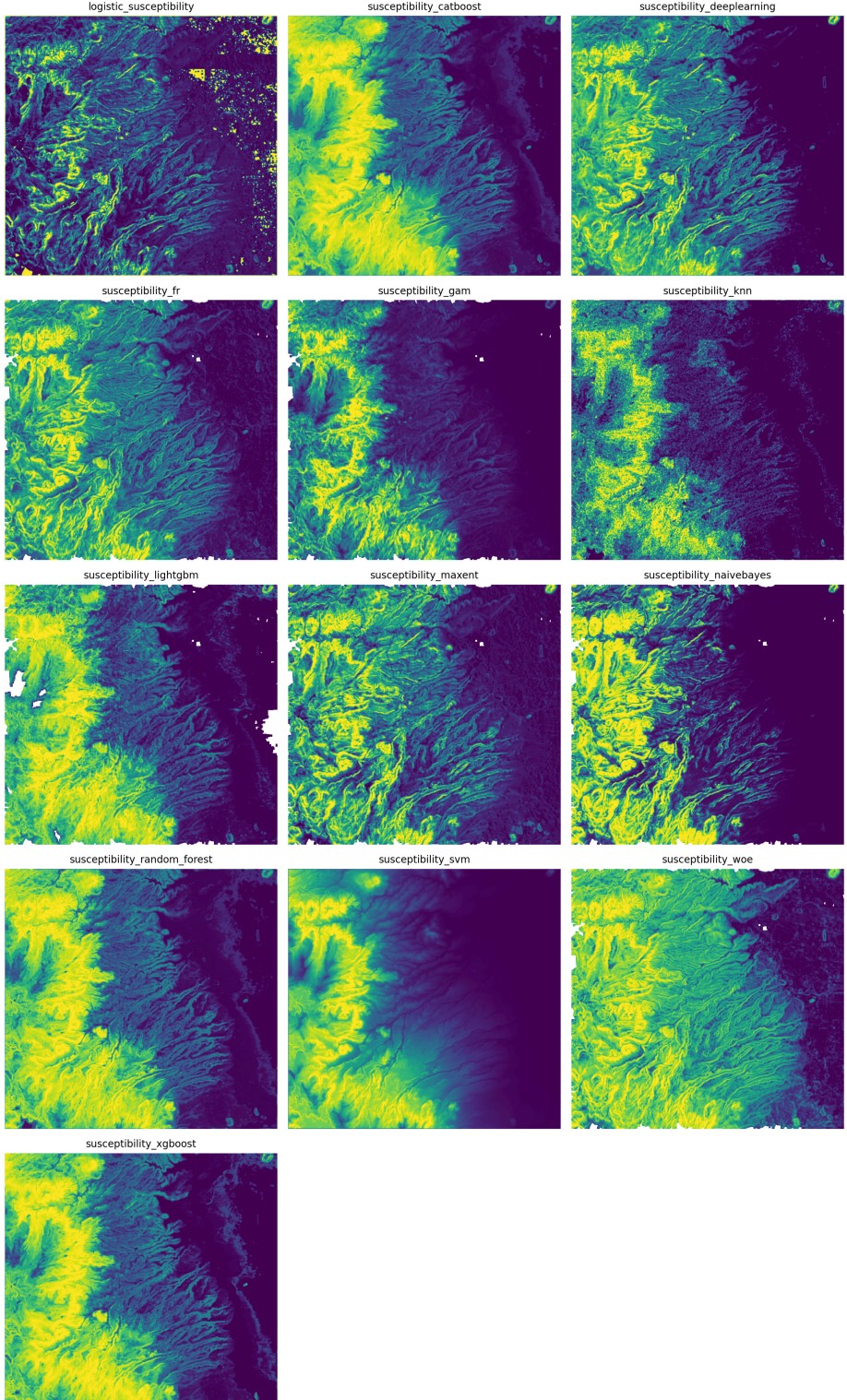

**Fig 5. Landslide susceptibility maps created by each tested model.** Matrix plot allows for quick comparison between susceptibility maps.

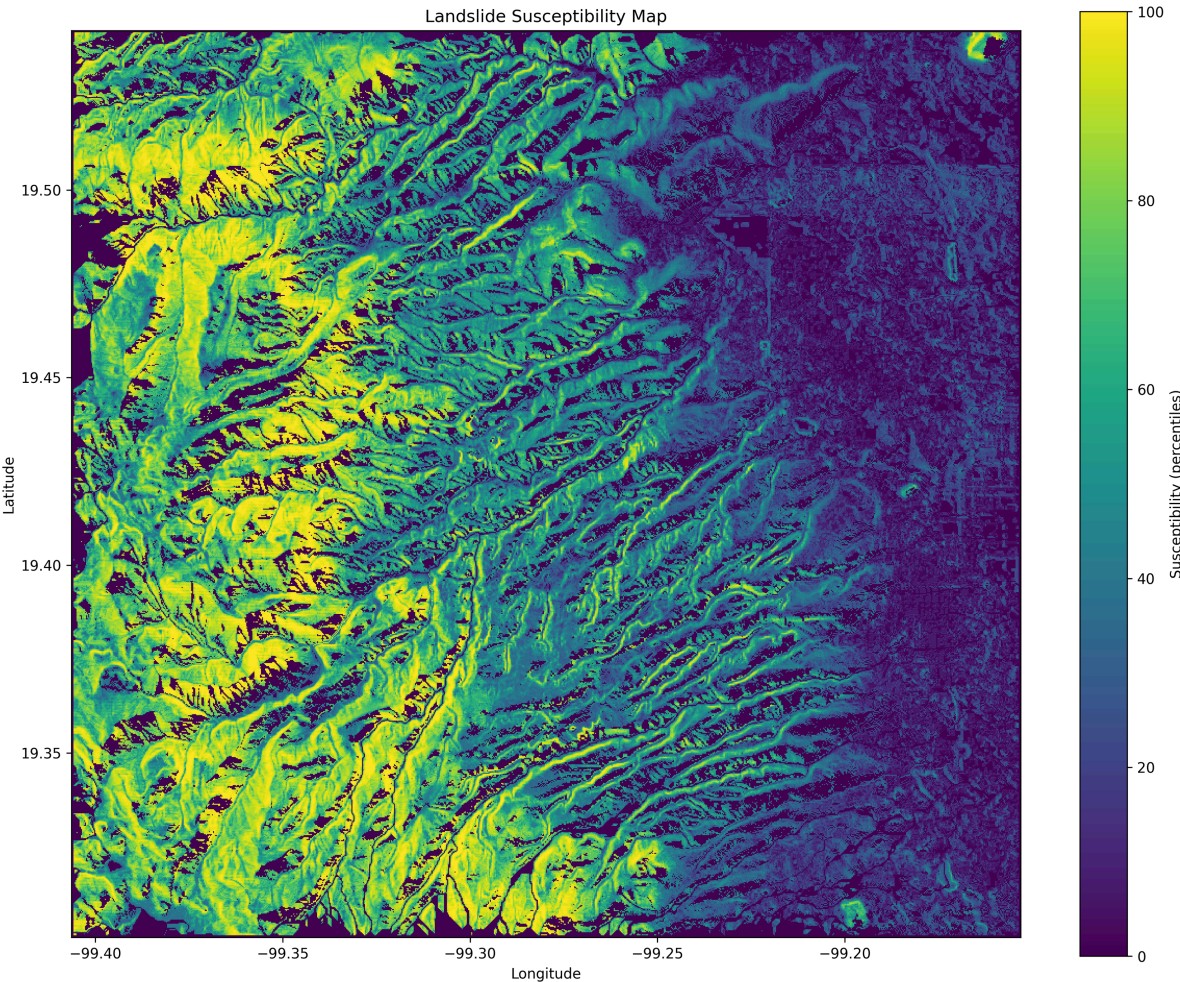

**Fig 6**. **Raw susceptibility map generated with XGBoost.** Full high-resolution susceptibility map of the champion model.

The National Center for Disaster Prevention in México (CENAPRED) has elaborated a susceptibility map for the whole country. Although the public version is not available as a GeoTIFF—and consequently we were not able to exactly replicate the bounding box coordinates of our study region—we present this approximate version to compare with our XGBoost output. Fig 9 shows the CENAPRED map and the resemblance with ours is clear. There exists an important amount of concordance in the Cuajimalpa, Álvaro Obregón, and Magdalena Contreras prefectures. Specifically, CENAPRED's map as well as ours, highlight the ravines on the east foothills to the west of Mexico City as high or moderate risk, a trait that is clearer when compared with our categorized susceptibility map presented at the end of the section. It is worth noting that our categorization methodology—that is, the susceptibility ranges used to define high or medium susceptibility—does not necessarily coincide with those used by CENAPRED.

To deepen the interpretation of our results, and given that the champion model was the XGBoost, we complemented its outputs with three additional approaches: Feature Importance (Gain), Information Gain Ratio, and the GeoDetector q-statistic, which are shown in Fig 10. The gain-based feature importance from XGBoost quantifies how influential an input is in the construction of the decision trees—it corresponds to the average gain in purity that a given input provides [64]—and highlights elevation and roughness as the primary contributors to model improvement across decision trees.

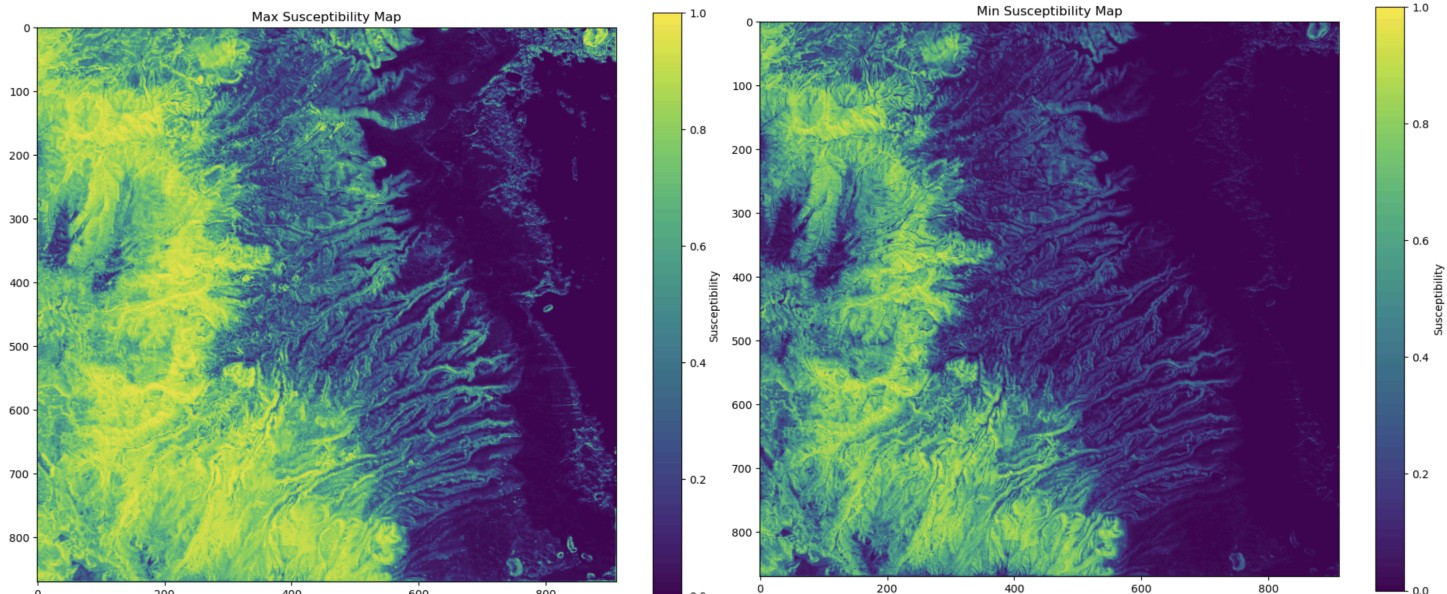

**Fig 7**. **Uncertainty quantification plots.** Maximum and minimum plots per areal unit as uncertainty quantification method, to portray the whole range of values that may arise.

**Table 6**. **Classification of landslide susceptibility values into categorical classes.**

| Category | Range of Susceptibility |
|---|---|
| Very Low Susceptibility | $0.00 \leq S < 0.20$ |
| Low Susceptibility | $0.20 \leq S < 0.40$ |
| Moderate Susceptibility | $0.40 \leq S < 0.50$ |
| High Susceptibility | $0.50 \leq S < 0.65$ |
| Very High Susceptibility | $0.65 \leq S \leq 1.00$ |

**Table notes:** Ranges help quickly identify riskier areas by emphasizing certain ranges. In our case, we used asymmetric ranges to call attention to specific areas.

The information gain ratio, although not natively part of the XGBoost suite and therefore calculated separately, evaluates how well features reduce uncertainty in the response by measuring the reduction in entropy when the dataset is split on a given input [64].

In our case, the Information Gain Ratio emphasizes the following variables as those that maximize predictive power: Elevation, Land Cover, D8 Slope, Roughness, Dinf Slope, and Slope, with a noticeable presence of TauDEM-derived variables. In parallel, the GeoDetector q-statistic, which is a spatial variance decomposition technique that quantifies the extent to which a feature spatially stratifies the variance of the dependent variable [64], ranks Elevation, Roughness, Slope, Dinf Slope, and D8 Slope as the most relevant variables. Notably, features related to topography and hydrological flow—particularly Slope-derived metrics and Elevation—emerge consistently across all three methods, underlining their central role in explaining landslide susceptibility and highlighting the importance of including these hydrological inputs.

To associate AGEB territorial divisions with landslide susceptibility values, we overlaid each AGEB polygon on the susceptibility map. Then, we assigned a representative susceptibility value to each AGEB. Nevertheless, a particular AGEB might have pixels with very different values. For instance, it was common to find AGEBs with a pixel of landslide susceptibility value of 0.075 and also another pixel with a value of 0.80 within that same AGEB, that is, widespread heterogeneity

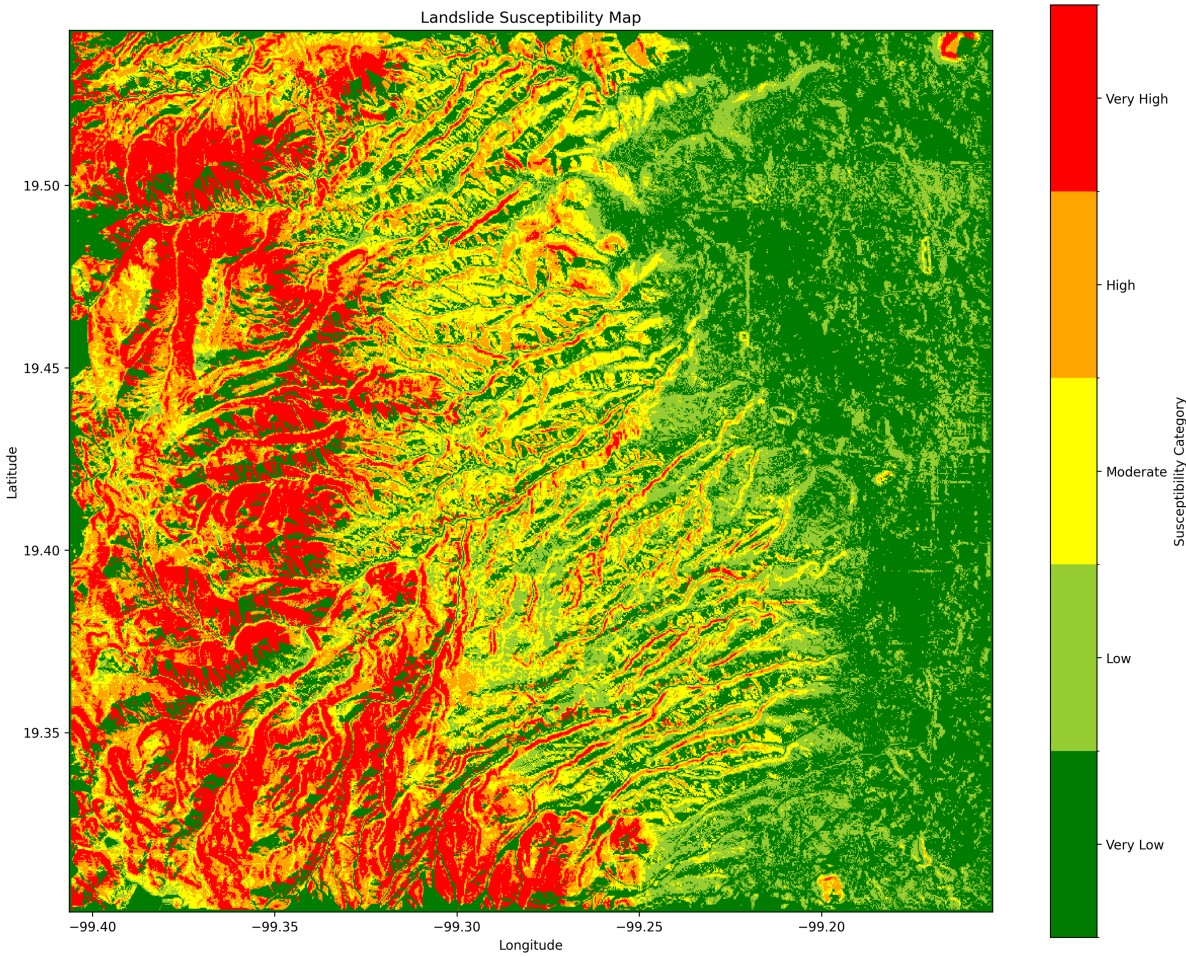

**Fig 8**. **Categorized susceptibility map.** Susceptibility values categorized to create a simpler-to-read map.

within AGEBs was not uncommon. This is because when the AGEB territorial division was created, landslide susceptibility was not taken into consideration. Consequently, within each AGEB, if a section of that AGEB lies in a leveled surface, and the other section lies in a steep hill, landslide susceptibility values within that AGEB could be very different. Therefore, to select a representative landslide susceptibility value for each AGEB, we took the 85th susceptibility percentile as a cutting point, that is, we say that for the i-th AGEB, its representative susceptibility value is $x$ if 85% of that AGEB values lie below $x$. For example, we say an AGEB has a representative value of 0.8 if 85% of the susceptibility values within that AGEB are less than or equal to 0.8.

After associating each AGEB with a representative value, we separated AGEBs into deciles according to their representative value. For example, AGEBs with susceptibility within a range value of 0 to 9.99 were put into the first decile category; those that have a susceptibility value within 10 to 19.99 are placed in the second decile category, and so on. This strategy created 10 susceptibility maps showing AGEBs with different susceptibility levels as portrayed in Fig 11, which highlights how AGEBs tend to move higher into the mountains, which are located to the west, as percentile susceptibility increases.

Furthermore, an important fact to clarify is that AGEBs lying in the city outside the western mountainous region naturally have low landslide susceptibility and tend to be low-income zones. As the mountain zone begins from east to west,

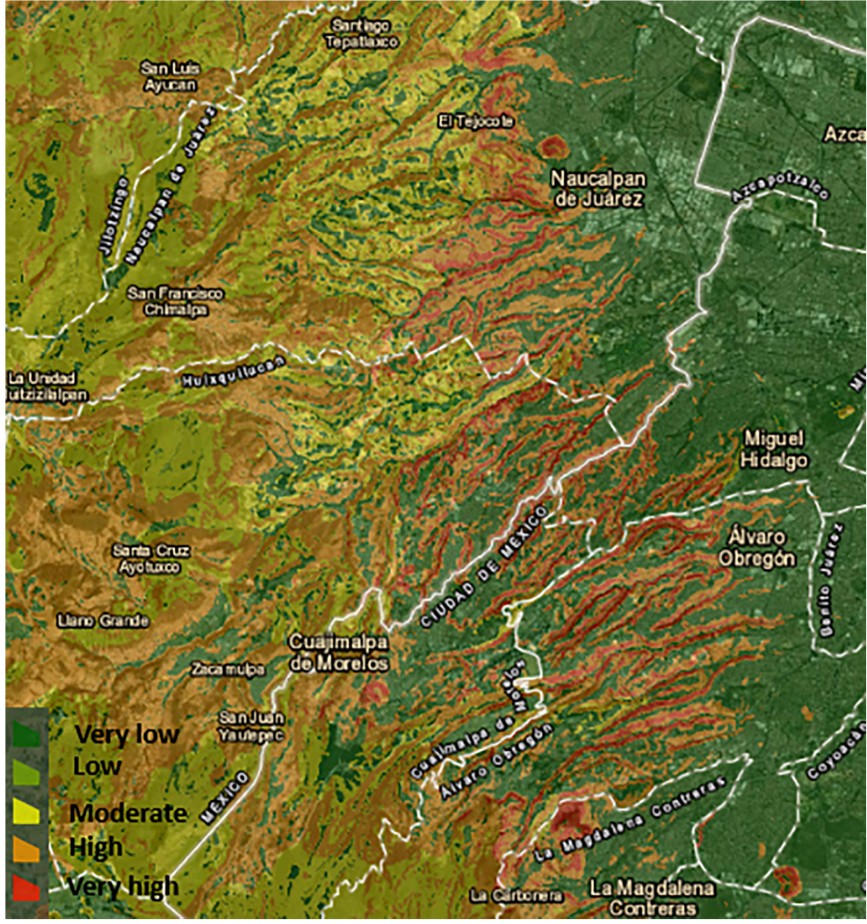

**Fig 9**. **CENAPRED official susceptibility map.** Notice that CENAPRED categorization values do not necessarily coincide with our methodology, since we intend to highlight different zones. This implies that the range and scope of our categorical map may differ from that of CENAPRED, yet this map is presented for transparency purposes.

within the mountain foothills, there is a low-income zone with relatively low landslide susceptibility—no more than 40%. Nevertheless, this zone also overlaps with an area of several ravines. The fact that these AGEBs are not assigned a higher susceptibility value will be addressed later. It is important to note that ravines typically occupy a relatively small proportion of an AGEB's total area. Since we assigned each AGEB the 85th percentile of its susceptibility value as a representative AGEB's susceptibility figure, this often resulted in a lower overall susceptibility score even in zones where ravines are not as big when compared with an AGEB's total area. In other words, AGEBs that contain small, yet highly hazardous zones like ravines, may still be assigned low susceptibility values given that the aggregation method assigned a representative susceptibility value—the 85th percentile—to the whole AGEB. Nevertheless, susceptibility maps—particularly our categorized version—clearly highlight these ravine zones as high-susceptibility areas.

Likewise, when landslide susceptibility starts at 40%, there is an interesting phenomenon: we found wealthy neighborhoods and low-income households almost side by side, which is a consequence of the aforementioned Santa Fé and Interlomas high-end residential, commercial, and office projects [12]. Finally, we merged the AGEB landslide susceptibility database with the Social Lag database we created, which is described in the Materials and Methods section. As a result, each AGEB was paired with susceptibility values, a Social Lag value, and the social variables with which the index was built. These are the inputs that will be needed in the next section.

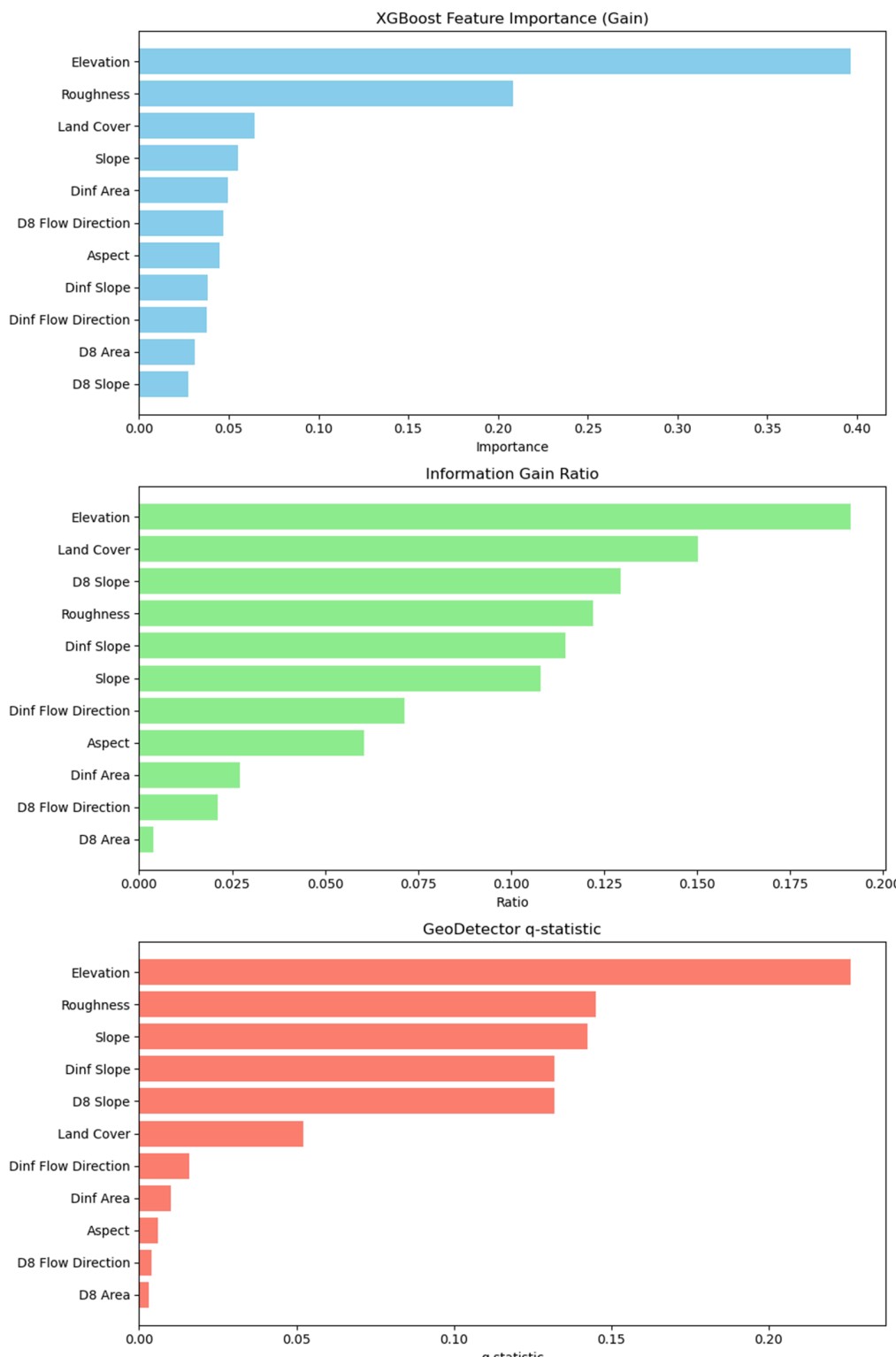

**Fig 10. Feature importance plots.** Feature importance, Importance gain, and Geodetector ranks features according to their importance.

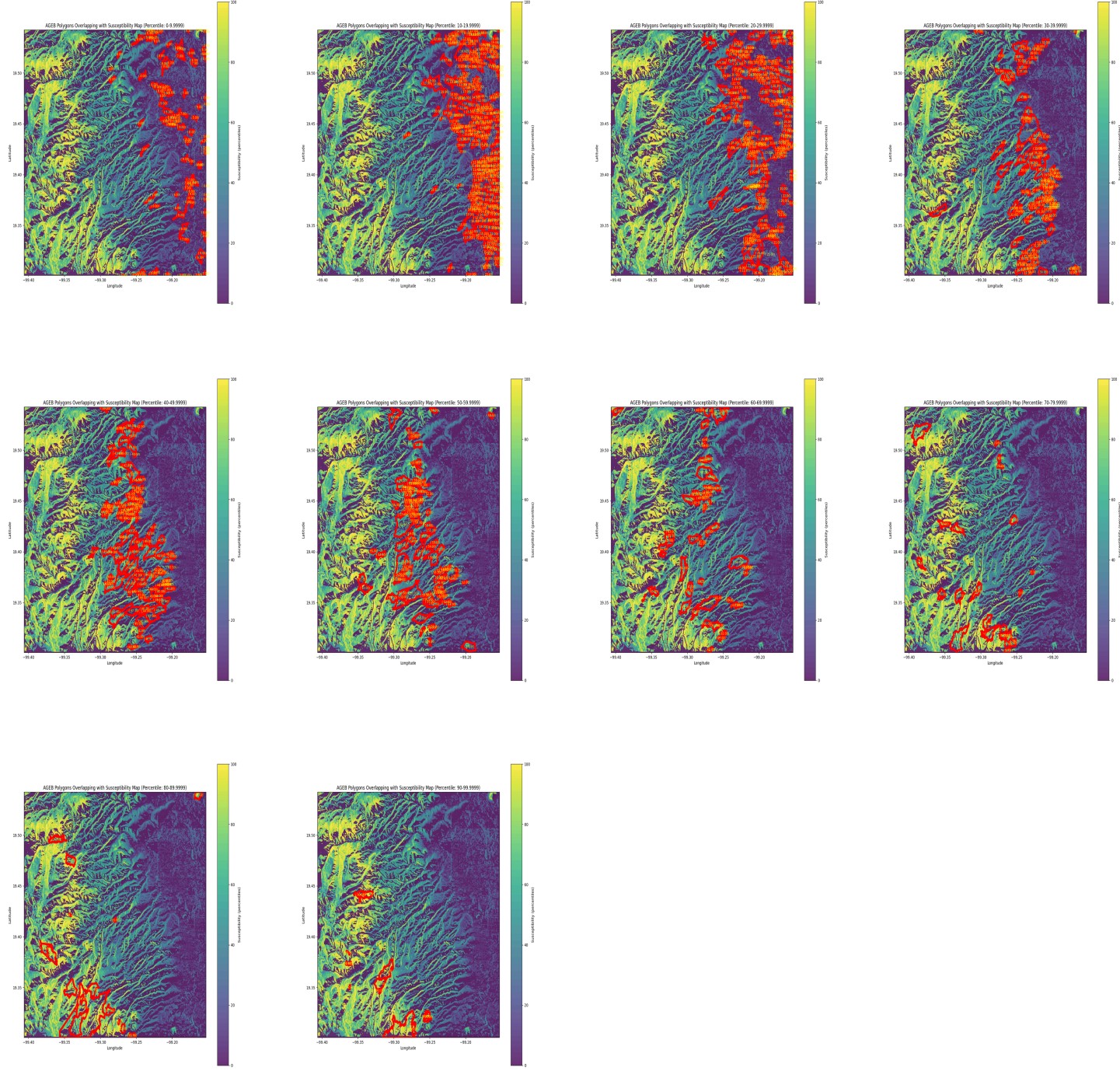

**Fig 11. Combined landslide susceptibility plots.** Susceptibility tends to increase to the west, therefore, as it increases, the corresponding AGEBs are located to the west.

## Association between percentile susceptibility and social lag

As mentioned before, one of the main goals of this study is to investigate whether a relationship exists between landslide susceptibility and social lag variables. In particular, we are interested in assessing whether higher levels of susceptibility

are associated with greater degrees of social lag. To this end, the rest of our analysis is structured in two parts. The first focuses on the relationship between landslide susceptibility and the Social Lag Index (SLI), and the second examines the association between landslide susceptibility and each of the individual variables with which the SLI was built.

To begin with, as shown in Fig 12, different susceptibility levels may correspond to different patterns of association—for instance, patterns observed in low-susceptibility areas may differ from those in medium- or high-susceptibility regions. Therefore, we conducted our analysis by partitioning the dataset to obtain a better understanding of its underlying characteristics. We begin by evaluating this relationship using the full dataset (without partitioning). Subsequently, to enhance the accuracy and resolution of our analysis, we applied a changepoint detection algorithm to get a more natural partition of the data into segments. Finally, we divided our data into deciles and examined their potential associations. For the association between landslide susceptibility and SLI, we used SHAP values and copula-based models to ensure robustness in the presence of nonlinear dependencies for each of the partitions obtained.

Once the partitioning method was selected, the analysis methodology that we applied was the same for each of the partitions mentioned before -whole dataset, changepoint partitioning, and decile partitioning. For each partition (and for the full dataset), the procedure was as follows: In a first stage, we took the numerically-continuous social lag index and landslide susceptibility values to run a Random Forest and Copula models to estimate the relationship magnitude and direction. As stated before, the way we addressed this dataset was by using SHAP values in the beeswarm form (hereafter referred to as SHAP plots) as an exploratory analysis for the direction and strength of the association between variables. To get the SHAP figures, we needed first to fit a machine learning model—in our case, this was the aforementioned Random Forest model. Estimating this model was also useful since we were able to get the $R^2$ value as a goodness-of-fit measure, which provides further support and credibility to the SHAP plots.

In a second stage, we estimated five copula models to get a robust association measure. We also used copula Kendall's $\tau$ as a standardize non-linear association metric. The fitted models were: Normal, T, Clayton, Gumbel, and

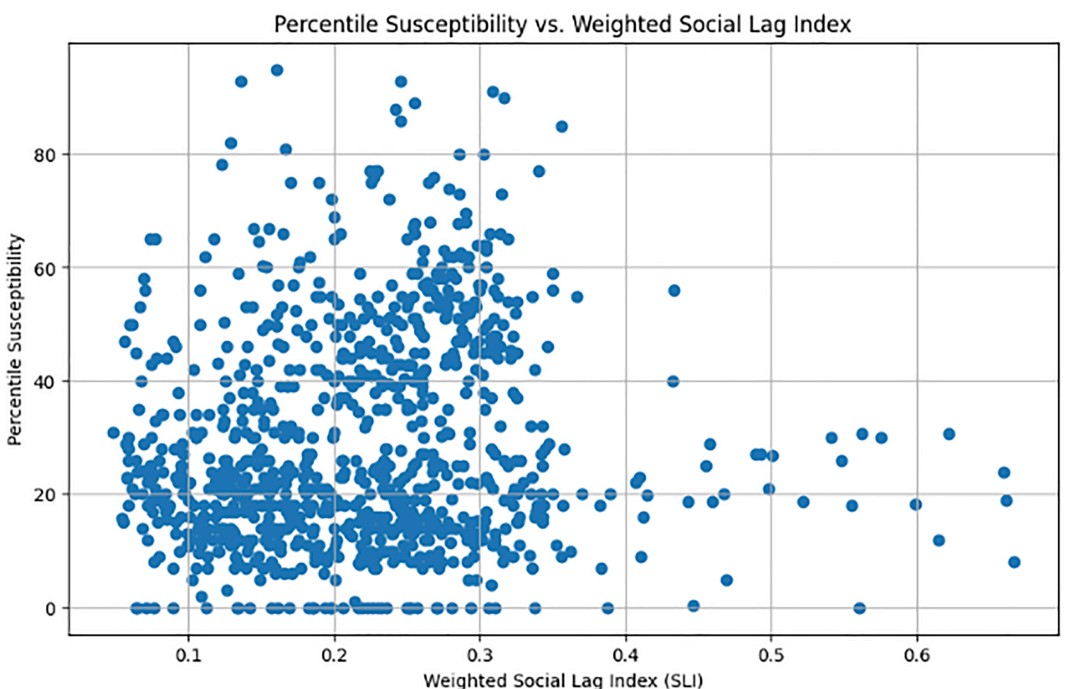

**Fig 12. SLI vs landslide susceptibility.** Social lag index plotted against different landslide susceptibility values.

Frank. For each of them, we calculated the log-likelihood, dependence parameter $\theta$, and Kendall's $\tau$. For each partitioning we chose the best copula model according to the highest log-likelihood value.

**Landslide susceptibility vs. SLI (full dataset).** Taking the whole dataset, that is, without decile-based or any other subdivision, the Random Forest model yielded an $R^2 = 0.8151$, which is a decent fit. It has a mean importance of 1.33, where mean importance measures the model's drop if SLI values were randomly shuffled. The fact that there is a 1.33 drop in $R^2$ performance implies that SLI is very important. Furthermore, Fig 13 shows the SHAP plot; it portrays the impact of SLI on landslide susceptibility predictions, with a color gradient denoting the feature value (blue = low SLI, red = high SLI).

In our case, Red dots lie predominantly on the positive side of SHAP values, which extend farther than values on the left, suggesting that when SLI is high, the model response is high as well; hence, higher social lag is associated with higher susceptibility predictions. On the negative SHAP side, low SLI values cluster closer to zero than the positive ones, and are mostly colored blue, suggesting that low SLI values are associated with lower model output on susceptibility. Hence, there are both zones with high and low SLI with high landslide susceptibility; however, in general terms and because of the longest right-red extension of the SHAP plot, we infer that overall disadvantaged communities, as measured by SLI, are more exposed to landslides.

As stated before, since our interest is to provide robust evidence of the relationship we are studying, we estimated copula models, which are well suited to dealing with nonlinear relationships. Results are shown in Table 7. According to the log-likelihood metric, the champion model is a Frank copula, which is symmetric and does not have upper or lower tail dependence. Kendall's $\tau$ has a value of 0.08993, implying a positive—although not necessarily strong—relationship between landslide susceptibility and SLI, providing further support for the SHAP values interpretation. This relationship is confirmed by the model dependence parameter, $\theta = 0.814727$, where positive $\theta$ values imply positive dependence structures. Given that $\theta$ parameter is not a standardized value, nothing should be inferred about the magnitude of the relationship. Nevertheless, and as just stated, since the value is positive, we can safely state that the higher the landslide susceptibility, the higher the SLI, further confirming the findings with the SHAP interpretation.

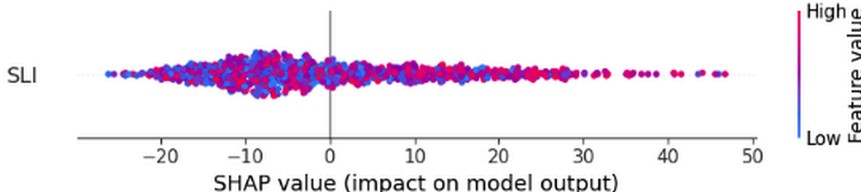

**Fig 13**. **Full dataset SHAP plot.** Beeswarm plot for the entire dataset.

**Table 7**. **Copula model performance based on log-likelihood and association.**

| Model | $\theta$ | Kendall's $\tau$ | Log-Likelihood |
|---|---|---|---|
| Frank | 0.8147 | 0.08993 | 68.264 |
| Gumbel | 1.0988 | 0.08993 | 58.857 |
| Normal | 0.1408 | 0.08993 | 57.635 |
| Clayton | 0.1976 | 0.08993 | 3.564 |
| Student-t | 0.1408 | 0.08993 | −36.329 |

**Table notes:** Kendall's $\tau$ is a rank-based dependence measure computed from the same pseudo-observations $(u,v)$ in every case therefore, $\tau$ is identical across copula families in this table. Copula families differ in their fitted parameters, tail dependence, and overall dependence shape, which is why log-likelihoods and theta coefficients may vary even when $\tau$ remains the same.

**Landslide susceptibility vs. SLI (changepoints).** In this setting, we allowed a changepoint algorithm to automatically detect and partition the dataset according to a penalty which, depending on the dataset structure, results in several different segments. In our case, the algorithm generated 4 segments. For each segment, we ran a random forest model and its associated SHAP plots; we also estimated copula models for each partition. Table 8 shows in the $R^2$ column that for every segment the model explains close to 80% of the variance. In addition, permutation importance values, since they are above 1 and with a small standard deviation, together indicate that the social lag index is an important input in each segment.

With respect to the association by segment, Fig 14 presents SHAP outcomes from which it can be inferred that the following patterns emerge:

- Segment 1, which corresponds to low-susceptibility AGEBs inside the city, SHAP values are centered around zero with both low and high SLI values present, indicating that social lag has a mixed influence on the model's output in this area, mainly because in these areas, there are several flat zones along with steep ravines near poor neighborhoods.
- Segment 2, covering a transition zone including affluent areas like Santa Fé and Interlomas, SHAP values are mostly negative, suggesting that both high and low SLI values contribute to lower predicted susceptibility. This reflects the socio-economic heterogeneity in this segment.
- Segment 3, a shift occurs: high SLI values (red) correspond to positive SHAP values, indicating that social lag contributes positively to predicted susceptibility. This implies a stronger relationship between social lag and landslide susceptibility.
- Segment 4 is particularly noteworthy, since there are two important stories developing: on one hand, high SLI values strongly contribute to high susceptibility (far right, red points), emphasizing vulnerability in highly marginalized areas. On the other hand, low SLI values also contribute positively to susceptibility (left side), suggesting that in mountainous regions, high landslide risk can occur even in socially advantaged AGEBs—likely due to topographic or environmental factors unrelated to social lag.

Finally, our copula analysis suggests that, based on Kendall's Tau, there is always a positive association between landslide susceptibility and SLI -to varying degrees- as shown in Table 9. Starting at the first segment, where we have a Tau of 0.0159. It is not surprising to find this weak association since overall this first segment is associated with low landslide susceptibility, and since the value associated with each AGEB does not necessarily reflect that there may be relatively smaller zones with very high susceptibility, such as ravines. The second segment shows a slightly weaker association, yet the third segment exhibits a substantial increase in the link between susceptibility and SLI; it is worth noting that this segment is associated with high landslide susceptibility. Finally, the last segment (with the highest susceptibility values) shows a weak association, but stronger than at the initial segments.

The explanation we offer about this outcome is as follows: first, the population in areas of high landslide susceptibility is actually low; secondly, the population living in areas of high landslide susceptibility is far from urban centers,

**Table 8. Segment-wise model performance and SHAP-based importance of the Social Lag Index (SLI).**

| Segment | Start | End | $R^2$ Score | Importance | Std. Dev. |
|---|---|---|---|---|---|
| 1 | 0 | 290 | 0.787 | 1.225 | 0.093 |
| 2 | 290 | 620 | 0.807 | 1.284 | 0.077 |
| 3 | 620 | 800 | 0.828 | 1.361 | 0.098 |
| 4 | 800 | 969 | 0.795 | 1.259 | 0.097 |

**Table notes:** The model maintains consistently high $R^2$ values across all segments, suggesting strong predictive performance of SLI on landslide susceptibility. Permutation importance values indicate SLI is the key driver in each segment, with low variance (standard deviation), reinforcing its influence.

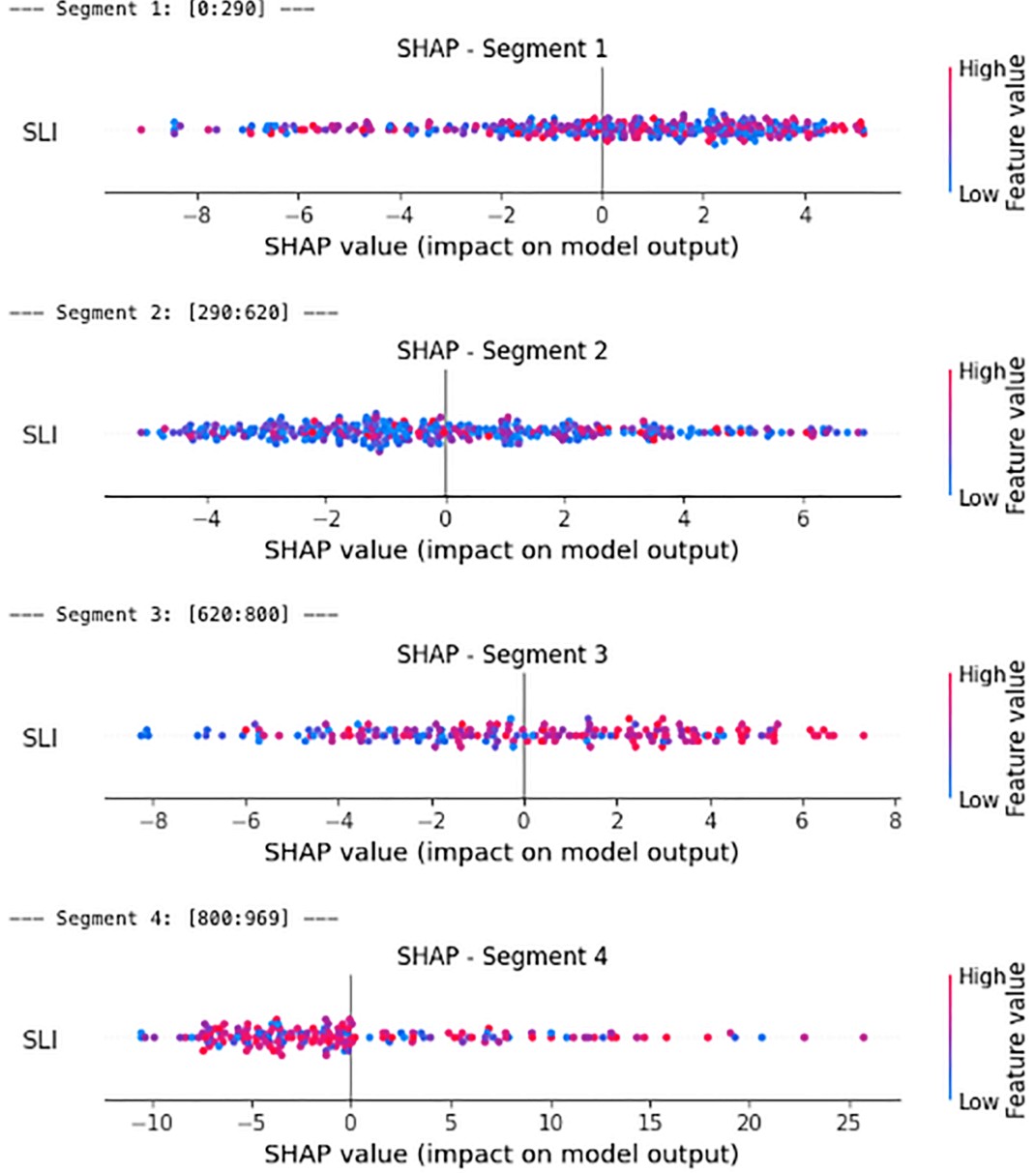

**Fig 14. SHAP plots for changepoint-induced partitions.** Beeswarm plot using changepoint induced partitioning.

therefore, these communities tend not to depend on services related to cities which, in a way, makes them self-sufficient communities, some of them in the form of *ejidos*, which not only are self-contained communities but also have this innate nature whereby the community members tend to support each other as much as possible. The very nature of these self-contained communities could be a factor in the reduced association that is found in extreme susceptibility zones.

**Landslide susceptibility vs. SLI by deciles.** In this section, we divided the dataset into deciles to obtain a more granular data review. To begin with, Table 10 shows that, on average, the random forest models fitted in each decile have a decent $R^2$ value close to 0.80. Furthermore, permutation importance coefficients are high in every segment, suggesting a noticeable association between the variables of interest, which, along with a reduced standard deviation—although

**Table 9.** Segment-wise copula model selection based on log-likelihood.

| Start | End | Best Copula | θ | Kendall's τ | Log-Likelihood |
|-------|-----|-------------|--------|-------------|----------------|
| 0 | 290 | Clayton | 0.0323 | 0.0159 | 2.332 |
| 290 | 620 | Frank | 0.0695 | 0.0076 | 1.367 |
| 620 | 800 | Frank | 1.8569 | 0.1996 | 4.424 |
| 800 | 969 | Normal | 0.0665 | 0.0423 | 1.166 |

**Table notes:** This table reports the best-fitting copula for each segment of the data based on log-likelihood maximization. Although Kendall's τ remains relatively low in most segments, the copula choice varies, reflecting local dependence structure differences.

**Table 10.** $R^2$ and permutation importance by decile.

| Decile | $R^2$ | Permutation Importance | Permutation STD |
|--------|----------|------------------------|-----------------|
| 1 | 0.772940 | 1.149482 | 0.122063 |
| 2 | 0.764483 | 1.178920 | 0.139973 |
| 3 | 0.804075 | 1.238992 | 0.103111 |
| 4 | 0.784263 | 1.223580 | 0.130380 |
| 5 | 0.773395 | 1.210510 | 0.140828 |
| 6 | 0.838732 | 1.410190 | 0.176712 |
| 7 | 0.836098 | 1.389542 | 0.169988 |
| 8 | 0.833028 | 1.389455 | 0.168143 |
| 9 | 0.838510 | 1.375273 | 0.140492 |
| 10 | 0.816736 | 1.336401 | 0.138246 |

**Table notes:** This table summarizes the model performance and SLI importance within each susceptibility decile.

slightly higher than the values observed with changepoint partitions—reinforces the influence of SLI on landslide susceptibility.

With respect to SHAP values, Fig 15 shows that there is no important association in the first three deciles, which is to be expected, since susceptibility values below 30% do not commonly produce landslides and correspond to zones located mostly inside the city and usually away from foothills or any mountainous formation. Nevertheless, an interesting pattern emerges in deciles 4, 5, and 6, for which most of the points are associated with negative SHAP values, indicating that regardless of the susceptibility levels within the decile, social lag seems to be low. This coincides with the fact that these deciles are located within the very affluent and high-end Santa Fé and Interlomas prefectures. Decile 7 is a transition zone since it begins to show red dots, although the really interesting values lie in deciles 8 and 9. These deciles correspond to zones above Santa Fé and Interlomas, which are also far from the city. In these areas, high social lag values are associated with very high landslide susceptibility. Finally, for decile 10 low SLI values are associated with high landslide susceptibility, indicating that not all highly susceptible areas are socially disadvantaged under the SLI definition.

With respect to copula association, the high decile-based granularity makes it possible to observe the substantial variability present in the dataset. Outcomes in Table 11 confirm the previously shown results: lower deciles have reduced association as measured by Kendall's τ, and some deciles even have negative association. Particularly, intermediate deciles have low or negative association because high-end zones lie in these deciles. Moreover, as expected, higher deciles show a considerably stronger association, which slowly decays as susceptibility gets higher—and as we move further into the mountains and away from the city.

## Association between percentile susceptibility and variables related to social lag

To provide an in-depth analysis of the association between landslide susceptibility and social lag, we next examine the individual association between each of the variables that comprise the social lag index and landslide susceptibility. The variables included in the social lag index are:

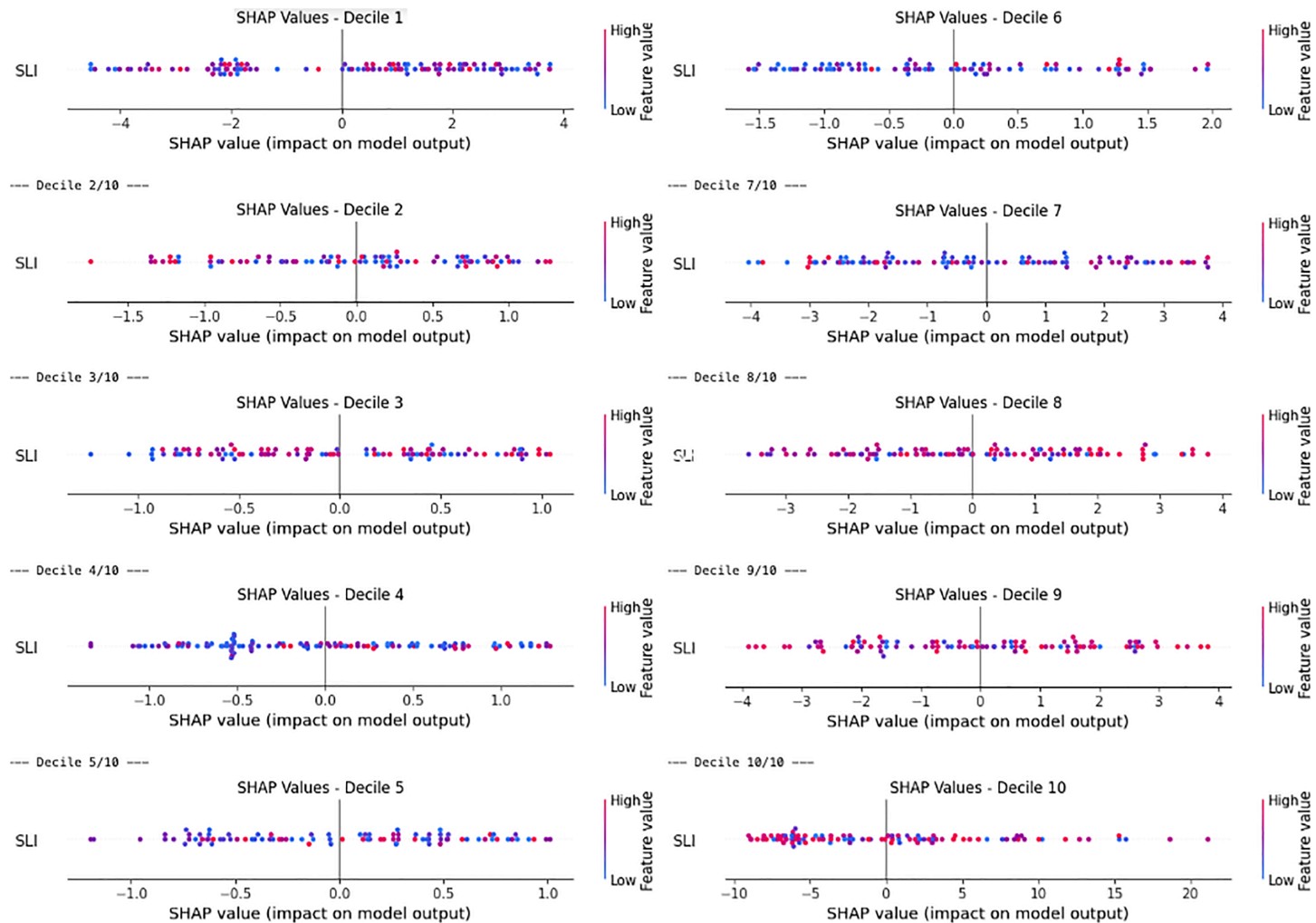

**Fig 15**. **SHAP plots for decile-induced partitions.** Beeswarm plot for each of the 10 partitions generated.

**Table 11**. Decile-wise copula model selection based on log-likelihood and dependence.

| Decile | Best Copula | $\theta$ | Kendall's $\tau$ | Log-Likelihood |
|---|---|---|---|---|
| 1 | Clayton | 0.1081 | 0.0513 | 3.264 |
| 2 | Normal | −0.0694 | −0.0442 | 2.873 |
| 3 | Frank | 0.9562 | 0.1053 | 1.263 |
| 4 | Clayton | 0.0607 | 0.0295 | 1.371 |
| 5 | Normal | −0.0638 | −0.0406 | 3.674 |
| 6 | Frank | 1.0916 | 0.1199 | 0.647 |
| 7 | Clayton | 0.2271 | 0.1020 | 1.859 |
| 8 | Frank | 0.8934 | 0.0985 | −0.037 |
| 9 | Gumbel | 1.0791 | 0.0733 | 4.523 |
| 10 | Frank | −0.1600 | −0.0178 | 0.561 |

**Table notes:** The table summarizes the best-fitting copula model per decile using log-likelihood as the selection metric. Most deciles exhibit weak monotonic dependence (low Kendall's $\tau$), but the sign and magnitude of the association vary across the susceptibility gradient.

- Var 1: Occupied Private Housing Units
- Var 2: Population Aged 15 and Older with Incomplete Basic Education
- Var 3: Population Aged 15–24 Not Attending School
- Var 4: Population Without Access to Health Services
- Var 5: People Living in Overcrowding
- Var 6: Housing Units Without a Toilet or Sanitary Facilities
- Var 7: Housing Units Without a Washing Machine
- Var 8: Housing Units Without a Refrigerator
- Var 9: Housing Units Without a Landline Phone
- Var 10: Population Aged 15 and Older Who Are Illiterate
- Var 11: Population Aged 6–14 Not Attending School
- Var 12: Housing Units With Dirt Floors
- Var 13: Housing Units Without Piped Water from the Public Network
- Var 14: Housing Units Without Drainage
- Var 15: Housing Units Without Electricity

**Exploratory data analysis.** To begin the exploratory analysis of the social variables used to construct the social lag index, we show boxplots for each variable in Fig 16. As it is fairly easy to appreciate, most of the variables contain an important number of outliers. Furthermore, housing units without drainage, housing units without piped water from the public network, and housing units with dirt floors stand out as being populated with highly dispersed data.

As an additional exploratory plot, Fig 17 presents population-related variables standardized by total population within each susceptibility class. It is important to note that this plot displays average values. In the first row, variables related to education and overcrowding exhibit disproportionately high values in the top two deciles, suggesting that higher landslide susceptibility is associated with worse educational and overcrowding conditions. This could be explained by the fact that high-susceptibility areas are typically located in mountainous regions far from urban centers, where access to schools and educational resources is limited.

In the second row, "Housing units without toilets" stands out with high values in the last two deciles. Interestingly, "Illiterate population aged 15 or older" exhibits extremely high values in the third decile. In the third row, no variable consistently stands out with high values in the upper deciles. However, "Housing units without refrigerators" shows somewhat elevated values, and "Mean population aged 6–14 not attending school" is noticeably lower in the last decile compared to the others.

Finally, in the last row, "Housing units with dirt floors", "Housing units without drainage", and "Housing units without electricity" exhibit considerably high values in the top two deciles. Notably, some variables, such as "Housing units without electricity", "Housing units without drainage", and "Housing units without piped water", also show high values in the lower deciles. This indicates that areas with lower susceptibility values, typically closer to urban centers, may paradoxically exhibit worse social conditions in some respects.

Having concluded this small exploratory introduction and with respect to robust modeling, we started with a Random Forest model to determine feature importance for each of the regressors, which are the variables with which SLI was constructed. We then computed the SHAP plots against landslide susceptibility to get a first glimpse of each variable's magnitude and direction. Finally, a copula model was fitted to each of the variables to obtain a confirmatory and more robust estimate of the magnitude and direction of association.

For the Random Forest model, we obtained an $R^2 = 0.9755$, a $MSE = 1.6476$, and an $MAE = 0.9$. With respect to individual variable-importance, Table 12 reports the weights. There is a soft decay from the most important variable to the next, and most variables played a relatively important role—only Occupied Housing Units is relatively unimportant. The most important variable is Housing Units Without a Landline Phone, followed by Population Aged 15–24 Not Attending

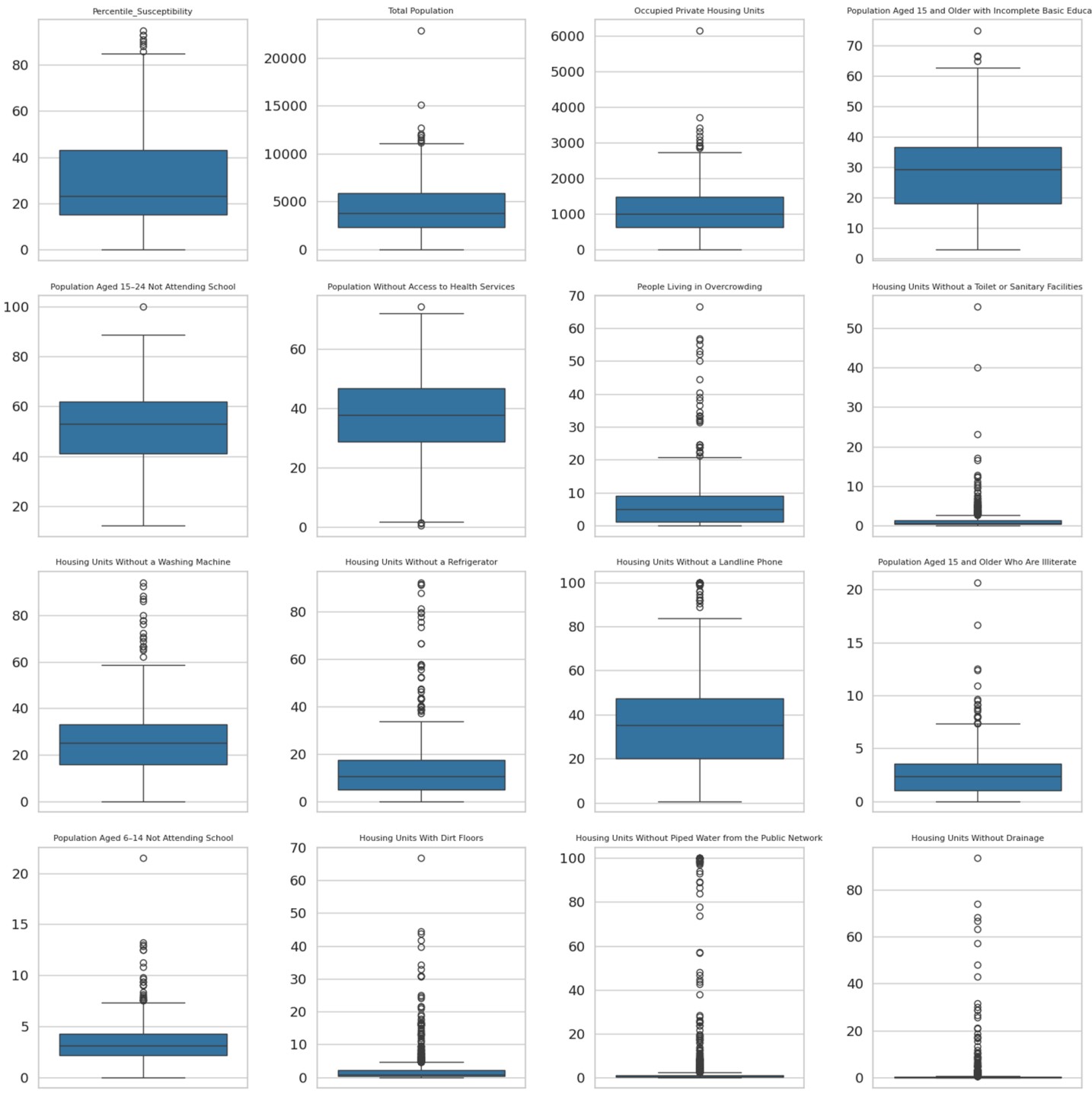

**Fig 16. Individual variable boxplots.** Boxplots portray first quartile, median, third quartile, and outlier data.

School, Housing Units With Dirt Floors, Housing Units Without a Washing Machine, and Housing Units Without a Refrigerator. Therefore, the most relevant variables are related to a severe lack of basic equipment in housing units, and with a few exceptions, most variables related to educational attainment are among the least important for the model.

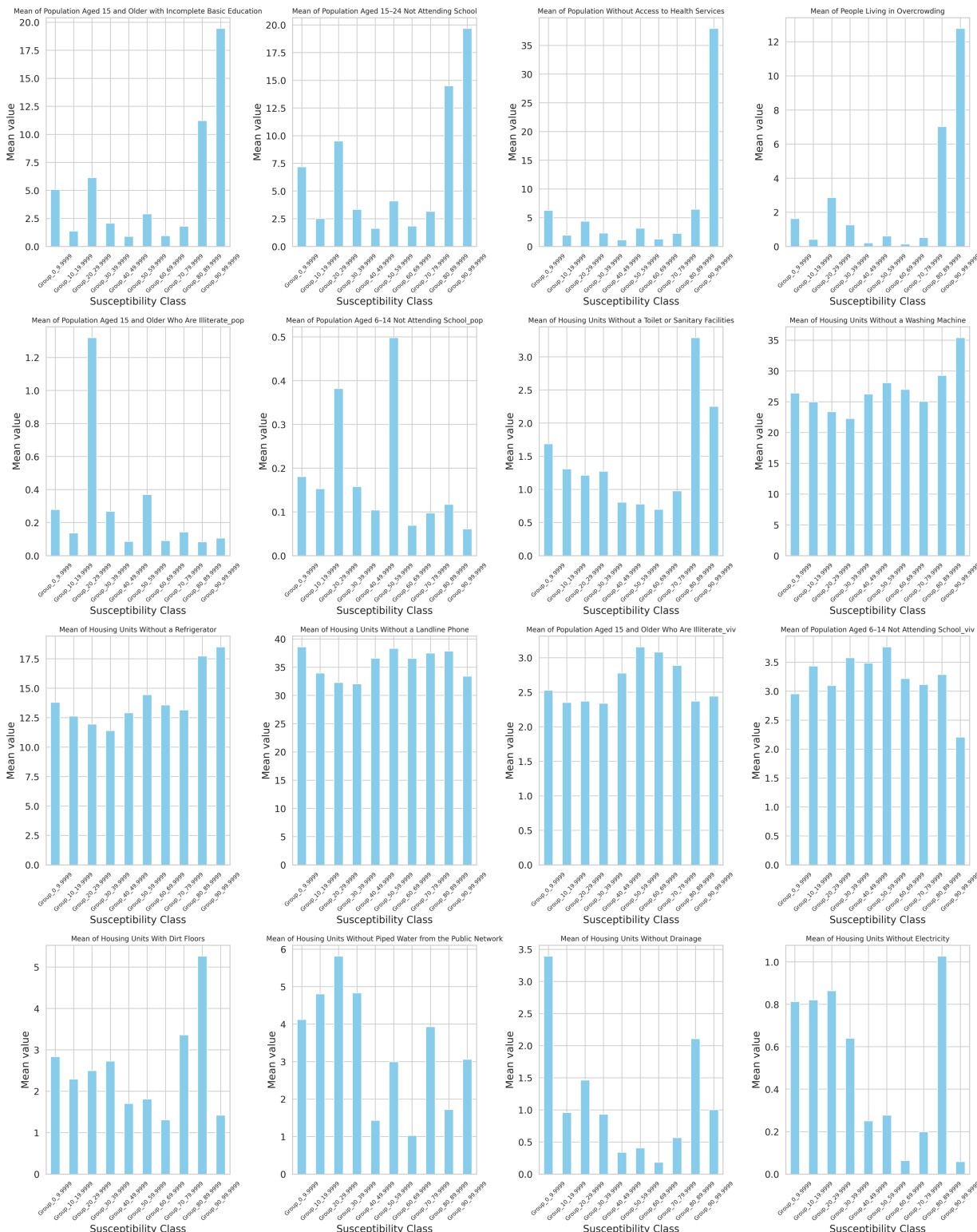

**Fig 17. Individual variable barplot.** Barplots shows the mean for each considered variable as separated by susceptibility class.

**Table 12**. **Feature importances from XGBoost model.**

| Feature | Importance |
|---|---|
| Housing Units Without a Landline Phone | 0.1320 |
| Population Aged 15–24 Not Attending School | 0.1006 |
| Housing Units With Dirt Floors | 0.0815 |
| Housing Units Without a Washing Machine | 0.0764 |
| Housing Units Without a Refrigerator | 0.0731 |
| Housing Units Without Drainage | 0.0694 |
| Housing Units Without a Toilet or Sanitary Facilities | 0.0642 |
| Population Without Access to Health Services | 0.0631 |
| Population Aged 15 and Older Who Are Illiterate | 0.0597 |
| Housing Units Without Piped Water from the Public Network | 0.0566 |
| People Living in Overcrowding | 0.0560 |
| Housing Units Without Electricity | 0.0491 |
| Population Aged 6–14 Not Attending School | 0.0489 |
| Population Aged 15 and Older with Incomplete Basic Education | 0.0431 |
| Occupied Private Housing Units | 0.0262 |

**Table notes:** Importance values indicate each feature's relative contribution to the predictive power of the XGBoost model for landslide susceptibility. Higher values reflect greater influence in model decisions.

Furthermore, Fig 18 presents SHAP values for all social vulnerability variables used in the landslide susceptibility model. The most influential variable is Population Aged 15–24 Not Attending School, whose high values (in red) are predominantly associated with positive SHAP values—indicating that in areas where more adolescents are not attending school, the model predicts higher landslide susceptibility. This suggests a strong link between educational exclusion and exposure to hazardous environments.

Other variables with similar positive associations include People Living in Overcrowding, Population Aged 15 and Older with Incomplete Basic Education, Occupied Private Housing Units, and Population Aged 15 and Older Who Are Illiterate. Together, these variables highlight how social deprivation, particularly related to education and housing density, is strongly associated with higher landslide susceptibility.

Finally, we found blue (low-value) points on the right-hand side of the SHAP axis (positive SHAP values) for variables such as Housing Units Without a Washing Machine, Housing Units With Dirt Floor, Housing Units Without Toilet or Sanitary Facilities, and Housing Units Without Drainage. In essence, this means that variables with high values of their own (e.g., a large number of housing units without drainage) are related to low landslide susceptibility. Notice that these variables can be framed around basic housing infrastructure; hence, as we mentioned in the previous sections, there are zones exposed to significant landslide susceptibility yet they are not necessarily simultaneously exposed to social lag as considered by basic housing infrastruture.

With respect to the association via Copula models, Table 13 shows the estimation results for each of the variables. The Best Copula column reports the champion model among those tested: Normal, Student-T, Clayton, Gumbel, and Frank, and model discrimination was made using the log-likelihood (shown in the last column and presenting only the highest value for each variable). The variables with the highest positive association according to Kendall's Tau are similar to those highlighted by the SHAP plots; that is, they refer to variables related mainly to poor educational attainment, overcrowding, and lack of access to healthcare. Specifically, these variables are: Population Aged 15–24 Not Attending School, Population Aged 15 and Older With Incomplete Basic Education, People Living in Overcrowding, Population Without Access to Health Services, and Population Aged 15 and Older Who Are Illiterate.

A second group of associations with less intensity is given by Housing Units Without Refrigerator, Washing Machine, and Landline Phone, which together with Population Aged 6–14 Not Attending School are also associated with high landslide susceptibility. To some extent, this group can be considered as linked to the absence of basic housing infrastructure.

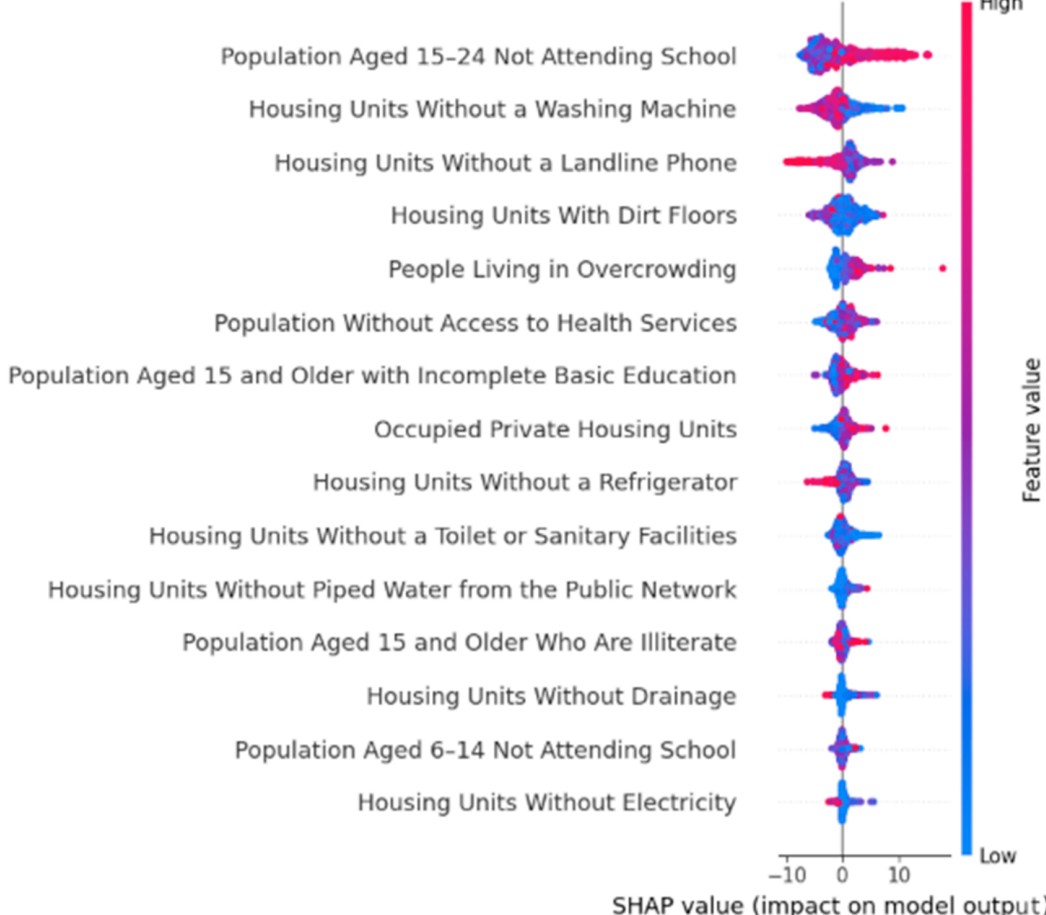

**Fig 18. SHAP plots for each variable that was used to construct the Social Lag Index.** Beeswarm plots showing potential association for individual social variables.

Finally, a smaller third group is comprised of variables with negative association; these are: Housing Units Without a Toilet or Sanitary Facilities and Housing Units Without Electricity.

To sum this complete results section, we have that:

- **Whole sample:** SHAP analysis indicates a positive but weak SLI–susceptibility association; Frank–Kendall's $\tau$ is small yet positive, plausibly due to spatial heterogeneity.
- **Changepoints (4 segments):**
    1. *Segment 1:* Positive SLI SHAP alongside both low and high susceptibility, reflecting co-existence of ravines and nearby flats under disadvantage.
    2. *Segment 2:* Low susceptibility largely independent of SLI (Santa Fé/Interlomas), with few high-SLI outliers.
    3. *Segment 3:* Clear co-occurrence of high SLI and high susceptibility in higher-altitude, peripheral zones.
    4. *Segment 4:* High susceptibility with both low and high SLI SHAP; self-contained communities (incl.ejidos) show relative social resilience despite exposure.
- **Deciles:** 1–2 low/negative association (intra-urban disadvantaged pockets); 3 strong (ravine concentration); 4–5 weak (affluent areas); 6–7 high (peripheral, exposed, not self-sufficient); 8–10 low/negative (peripheral, more autonomous).

**Table 13.** Copula fit per variable (variable IDs from master list).

| Variable | Best Copula | θ | Kendall's $\tau$ | Log-Likelihood |
|---|---|---|---|---|
| Var 1 | Frank | 0.3266 | 0.0363 | 38.57 |
| Var 2 | Frank | 1.0455 | 0.1149 | 65.71 |
| Var 3 | Gumbel | 1.1418 | 0.1242 | 39.62 |
| Var 4 | Gumbel | 1.1080 | 0.0975 | 13.74 |
| Var 5 | Normal | 0.1611 | 0.1030 | 153.61 |
| Var 6 | Frank | −0.6806 | −0.0753 | −139.45 |
| Var 7 | Frank | 0.4187 | 0.0464 | 36.45 |
| Var 8 | Normal | 0.1030 | 0.0657 | 78.17 |
| Var 9 | Frank | 0.3986 | 0.0442 | 23.27 |
| Var 10 | Normal | 0.1447 | 0.0924 | 107.45 |
| Var 11 | Frank | 0.4234 | 0.0470 | 52.25 |
| Var 12 | Clayton | 0.0971 | 0.0463 | 123.72 |
| Var 13 | Clayton | 0.0583 | 0.0283 | 149.84 |
| Var 14 | Clayton | 0.0031 | 0.0015 | 20.75 |
| Var 15 | Frank | −0.0989 | −0.0110 | −18.71 |

**Table notes:** This table summarizes the best-fitting copula model for each socioeconomic variable, identified by "Var" IDs from the master list since variable names are too long to fit in this table along with the other columns of interest. Kendall's $\tau$ indicates monotonic dependence strength.

- **Variable-level:** Higher susceptibility aligns most with educational deprivation, followed by limited healthcare access and overcrowding; secondary links to basic housing deficits (earthen floors, no landline, no washer/refrigerator).
- **Implication:** Where high susceptibility overlaps high SLI and low social mobility, risk is compounded; migrants and low-education groups appear especially exposed according to evidence.

## Discussion

Overall, we were able to identify a positive relationship between landslide susceptibility and social lag in many instances. This finding allows us to reconnect this work with our initial discussion: at the outset, we addressed the issue of people who migrate from other states to the nearby Mexico City areas, looking for a better life. We also stated that an increasing number of people who were living inside the city are moving to the city outskirts, because it is harder to get affordable housing inside Mexico City. With the obtained results, we have statistical evidence to suggest that there exists a social mobility trap in the context of landslides, that is, the more people move to the outskirts of the city on the western periphery, seeking a better life by being close to the industrial zones where there are jobs and where housing is affordable, the more these already vulnerable individuals are exposed to a higher landslide risk. This is particularly true in the ravine zones near the west foothills. Furthermore, not every person currently living in these zones is a migrant; many were born in the area, and they simply were born to an invisible yet ever-present landslide risk.

Decile analysis, as well as a changepoint detection algorithm along with copulas and SHAP values, together confirmed that the zone where the wealthy enclaves of Santa Fe and Interlomas are located have a considerable range of high susceptibility values. Furthermore, within this same zone, we found low-income enclaves side-by-side the newest and most luxurious buildings in the zone.

This fact highlights a dual story of high inequality: wealthy enclaves possess enough resources to build safely and resiliently even in risky zones; they have better resources to conduct sub-soil studies and to lay deep and safe foundations for buildings. They also have the resources to hire highly qualified professionals to construct their buildings, and then enjoy the amazing views offered in mountainous areas. In contrast, low-income enclaves do not have the same resources to build and to study the subsoil. Additionally, low-income households frequently build even without the supervision of a professional architect or structural engineer, and they even build without the authorities' formal knowledge and/or permission. This occurs because of a lack of supervision from authorities.

The fact that people move to landslide-prone areas might be psychological, at least in some aspects. According to decision theory, people's actions might be influenced by the so-called availability heuristic, by which people make decisions by overestimating the chances of events that rarely come to mind, such as landslides. This is because landslides are not necessarily a common event, that is, if a negative event does not occur frequently, that risk rarely intimidates those who might be exposed to it. Furthermore, the normalcy bias states that people underestimate the possibility of an emergency, thinking that such events cannot happen to them [65]. In this case, landslide events might have been underestimated.

Another important development refers to the inhabitants who are somehow excluded from society. They live high in the mountains in high susceptibility zones. We crudely say "excluded from society" because these enclaves lie so far from the city—some of them are not necessarily far, yet road access to these communities is so complicated that it is difficult to get in or out. Consequently, inhabitants need to base most of their activities in their places.

On one hand, this makes their communities self-sufficient in many respects, and the quality of their lives seems to be unaffected by being away from the city; indeed, they seem to be better off the way they are, as confirmed by visual inspection. On the other hand, the fact that they are so far from the city is noticed by the addressed social variables: schooling is severely low in almost every dimension measured, including high illiteracy, which is a very worrisome dimension; overcrowding is prevalent; access to healthcare is scarce, and lack of basic housing infrastructure is ever-present.

Additionally, well within the territory of these afar communities, there exists a type of social organization called *ejido*. These are self-contained shared communities in which lands, forests, and other resources are common to their inhabitants; they also share the economic activities and their profits [66]. The very nature of this self-supportive organization would make any social lag indicator mark these communities as overall low, since these communities intend to support each other, thus dissipating the nefarious effects of social lag. These communities exist in our study area well in the mountains, where high landslide susceptibility is to be found. There are at least five of these ejidos in Huixquilucan, eight in Naucalpan, two in Magdalena Contreras, and one in Cuajimalpa [67].

Policy implications are straightforward: our findings clearly associate vulnerable people with worrisome levels of landslide susceptibility. Therefore, people living in risky zones such as ravines should be warned of the imminent dangers; they should be advised about when to leave in case of heavy rain, and where they should go. What is more, a land-use change will not suffice as an incentive for people living in this zone to move out of risky areas. This is because they live there since they do not have a choice; instead, they need feasible alternatives where to move, they need affordable government financing to buy/develop housing outside the risk areas, and they need to stop being allowed to settle in risky zones.

Additionally, government officials must get closer to communities afar from the city in risky areas to provide schooling, healthcare, and affordable financing to acquire basic housing infrastructure. Authorities also need to provide legal protection, quality construction orientation, as well as any other basic benefits. Each segment of society in the study region is different, as portrayed by the decile and changepoint analysis; therefore, a differentiated policy should be applied to each section to meet specific needs.

Although a bit off the reach of this study, it is nevertheless necessary to mention that climate change will play an important role in the future of these disasters: heavier rains will foster increased pore-water pressure due to infiltration, making slopes far more unstable and potentially increasing the occurrence and frequency of earthflow slides.

In comparison to prior works which often relied on overlays or descriptive indices of social vulnerability, this study advances the literature in three ways. First, it explicitly quantifies nonlinear associations between landslide susceptibility and social lag through SHAP values, copula models, and segmentation techniques, thereby moving beyond correlation by visual overlay. Second, it integrates hydrological terrain factors derived from TauDEM into the analysis, ensuring that physical processes are realistically captured at fine spatial resolution. Third, it frames the susceptibility–lag nexus within a broader narrative of social mobility traps and behavioral risk perception, offering a conceptual link rarely addressed in the hazard literature. Taken together, these advances provide both methodological and theoretical contributions that set this

study apart and extend the global debate on how environmental hazards intersect with social lag and inequality in urban peripheries.

## Conclusion

We tested 13 machine learning models to construct a landslide-susceptibility map and used SHAP values and copulas to analyze their association with social lag in four complementary setups. While the citywide association is modest, change-point and decile partitions reveal regimes where high susceptibility and high social lag coincide in peripheral, steep terrain, and others where exposure is high but social conditions are relatively resilient. Limited schooling, poor access to healthcare, and inadequate housing emerge as the social dimensions most closely associated with higher susceptibility, pointing to increased vulnerability and constrained social mobility. TauDEM-derived hydrological variables rank among the most important predictors, highlighting the value of integrating hydrological modeling into susceptibility assessment and of tailoring policies to the distinct regimes identified.

## Limitations

The landslide inventory, which we obtained from the World Bank Global Landslide Catalog at ~1 km and resampled to 30 m, introduces uncertainty and likely under-represents small or remote events. Therefore, susceptibility in poorly monitored areas is probably conservative. Social conditions are captured through Social Lag Index which is aggregated at the AGEB level, which may smooth highly disadvantaged zones within otherwise "low-lag" units. As a consequence, our index remains subject to aggregation bias as any other index at AGEB aggregation level. A modest temporal mismatch between DEM, land cover and landslide dates, together with mean imputation of missing values and stratified subsampling, adds additional noise and implies that our susceptibility map is better acknowledged as a relative susceptibility ranking rather than an exact probability surface. Finally, our bootstrap uncertainty maps reflect variability in the XGBoost model due to sampling and training, but do not capture structural uncertainties from inventory incompleteness and temporal mismatch. Therefore, these residual bias, which include social data aggregation, should be considered when interpreting results.

## Future work

We used 15 variables to construct the social lag index to make it comparable to the original CONEVAL index. Nevertheless, there were about 250 available social variables in the database; therefore, the next natural step is to explore the complete dataset in search for other social lag dimensions. Additionally, we may include other conditioning factors in the landslide susceptibility models, such as distance to roads, NDVI, and distance to tectonic plates. Finally, it is of paramount importance to include climate projections to determine not only where risky landslide zones are, but also where they will be, as global climate becomes more extreme.

## Acknowledgments

We deeply thank David Gerzsenyi for his openness on sharing information.

## Author contributions

**Conceptualization:** Mario Alejandro Mercado Mendoza, Armando Sánchez Vargas, Pierre Mokondoko.

**Data curation:** Mario Alejandro Mercado Mendoza, Armando Sánchez Vargas, Pierre Mokondoko.

**Formal analysis:** Mario Alejandro Mercado Mendoza, Armando Sánchez Vargas, Pierre Mokondoko.

**Funding acquisition:** Mario Alejandro Mercado Mendoza.

**Investigation:** Mario Alejandro Mercado Mendoza, Armando Sánchez Vargas, Pierre Mokondoko.

**Methodology:** Mario Alejandro Mercado Mendoza, Armando Sánchez Vargas, Pierre Mokondoko.

**Project administration:** Mario Alejandro Mercado Mendoza.

**Resources:** Mario Alejandro Mercado Mendoza, Armando Sánchez Vargas, Pierre Mokondoko.

**Software:** Mario Alejandro Mercado Mendoza, Armando Sánchez Vargas, Pierre Mokondoko.

**Supervision:** Mario Alejandro Mercado Mendoza.

**Validation:** Mario Alejandro Mercado Mendoza, Armando Sánchez Vargas, Pierre Mokondoko.

**Visualization:** Mario Alejandro Mercado Mendoza, Armando Sánchez Vargas, Pierre Mokondoko.

**Writing – original draft:** Mario Alejandro Mercado Mendoza, Armando Sánchez Vargas, Pierre Mokondoko.

**Writing – review & editing:** Mario Alejandro Mercado Mendoza, Armando Sánchez Vargas, Pierre Mokondoko.

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
