## [Decision Letter · Decision Letter 0]

11 Mar 2025

PONE-D-25-00532Relationship between landslide susceptibility and social lag in Mexico City: the Case of the West PeripheryPLOS ONE

Dear Dr. Mercado Mendoza,

Thank you for submitting your manuscript to PLOS ONE. After careful consideration, we feel that it has merit but does not fully meet PLOS ONE’s publication criteria as it currently stands. Therefore, we invite you to submit a revised version of the manuscript that addresses the points raised during the review process.

We look forward to receiving your revised manuscript.

Kind regards,

Gayathiri Ekambaram, Ph.D

Academic Editor

PLOS ONE

Journal Requirements:

Additional Editor Comments:

The manuscript “Relationship between landslide susceptibility and social lag in Mexico City: The Case of the West Periphery” presents an interesting topic that integrates landslide susceptibility analysis with socio-economic factors. However, based on the two detailed reviews, there are significant methodological concerns that must be addressed before the manuscript can be reconsidered.

Key Issues to Address in the Revision:

1. Justification of the Frequency Ratio (FR) Method:

o Reviewer 1 suggests that the use of FR needs stronger justification, especially given the availability of more advanced machine learning techniques (e.g., Random Forest, SVM, CNN).

o A comparative analysis should be conducted between FR and an ML-based model to demonstrate its effectiveness.

2. Weak Statistical Model for Social Lag Analysis:

o The multinomial logistic regression model shows an extremely weak fit (pseudo R² = 0.007352), making the relationship between social lag and landslide susceptibility statistically weak.

o Alternative modeling techniques (e.g., feature selection, interaction terms, or different regression models) should be considered.

3. Validation of the Landslide Susceptibility Model:

o The lack of independent validation data is a major concern. It is crucial to include:

A validation dataset of past landslides to assess accuracy.

Performance metrics such as AUC-ROC curves to validate the predictive strength of the model.

4. Contradictions in Results and Hypothesis:

o The manuscript hypothesizes that higher social lag should correlate with higher landslide susceptibility, but findings show the opposite trend.

o A clearer explanation of this contradiction is required. If the hypothesis is incorrect, it should be revised with alternative theoretical support.

5. Choice of Huber Regression and Lack of Diagnostics:

o The rationale for using Huber regression should be justified over other robust regression techniques (e.g., Quantile Regression, Ridge Regression).

o The study lacks model diagnostics (e.g., residual analysis, VIF, or multicollinearity tests) to verify the robustness of Huber regression.

6. Missing Data Issues and Temporal Bias in Data Sources:

o There is no explanation of how missing values in social lag variables were handled.

o The temporal mismatch between datasets (e.g., 2007 DEM vs. 2020 social data) could introduce bias.

7. Comparison with Official Landslide Hazard Maps:

o Given that Mexico City already has existing landslide hazard maps, the manuscript should compare its susceptibility map with official hazard maps to assess its reliability.

8. Lack of Uncertainty Quantification:

o Landslide susceptibility models are probabilistic in nature. The study should provide confidence intervals for its estimates.

9. Inadequate Literature Citations and Reference Formatting Issues:

o Reviewer 2 highlights improper references, missing citations from high-impact peer-reviewed sources, and unstructured reference formatting.

o The literature review should incorporate recent and relevant citations, including key works on landslide susceptibility modeling.

10. Data Resolution and Spatial Analysis Issues:

• There is no explanation of how datasets with different spatial resolutions (e.g., Landsat, MODIS, and DEM data) were standardized.

• Image enhancement or fusion techniques may be required to ensure data consistency.

11. Missing Discussion:

• Sensitivity analysis should be performed to quantify the impact of different variables on landslide susceptibility.

• The manuscript lacks a proper discussion of computational efficiency, feature importance, and the limitations of the proposed method.

12. Both reviewers express concerns about language clarity, grammatical errors, and ambiguous phrasing. A thorough English language revision is necessary to ensure the manuscript is clear, coherent, and professionally structured.

Reviewers' comments:

Reviewer's Responses to Questions

**Comments to the Author**

1. Is the manuscript technically sound, and do the data support the conclusions?

Reviewer #1: Partly

Reviewer #2: No

2. Has the statistical analysis been performed appropriately and rigorously?

Reviewer #1: Yes

Reviewer #2: No

3. Have the authors made all data underlying the findings in their manuscript fully available?

Reviewer #1: Yes

Reviewer #2: No

4. Is the manuscript presented in an intelligible fashion and written in standard English?

Reviewer #1: No

Reviewer #2: No

5. Review Comments to the Author

Reviewer #1: 1. The authors claim that they chose the Frequency Ratio (FR) method due to its simplicity, interpretability, and its presence in previous literature. However, this is not a sufficient justification. Given the availability of more robust machine-learning techniques for susceptibility mapping, such as Random Forests (RF), Support Vector Machines (SVM), or Deep Learning (CNN), the manuscript should justify why FR was the best choice for this study.

2. The authors acknowledge that machine learning techniques are more advanced but dismiss them without testing or benchmarking them against FR. A comparative analysis between FR and ML methods should be included to validate this choice.

3. The study attempts to establish a relationship between social lag and landslide susceptibility using multinomial logistic regression. However:

i. The model's pseudo R² value is extremely low (0.007352), indicating an extremely weak fit.

ii. The p-values for most coefficients are non-significant, implying that the hypothesis of an association between landslide susceptibility and social lag is not supported.

iii. The authors acknowledge this limitation but do not propose a corrective measure, such as feature selection, interaction effects, or alternative modeling approaches.

4. The authors use Huber regression to study individual social lag components, citing outlier robustness as the main reason. However:

i. The rationale for choosing Huber regression over other robust techniques (e.g., Quantile Regression, Ridge/Lasso Regression) is missing.

ii. Collinearity issues among social lag variables are not addressed, potentially leading to misleading results.

iii. The paper lacks model diagnostics, such as residual plots or variance inflation factor (VIF) analysis, to confirm the robustness of Huber regression.

5. The study lacks an independent validation dataset of past landslides to assess the accuracy of the generated susceptibility map.

6. It is crucial to calculate performance metrics, such as the Area Under the Curve (AUC) of the Receiver Operating Characteristic (ROC) curve, for landslide susceptibility models.

7. Mexico City has various landslide hazard maps published by government agencies or research institutions.

8. The study should compare the proposed susceptibility map with official landslide hazard maps to assess its reliability.

9. The results section contradicts the initial hypothesis. The authors hypothesize that social lag should correlate with landslide susceptibility but later find that areas with high landslide susceptibility tend to have lower social lag.

10. This contradiction is not well explained, nor is an alternative hypothesis proposed.

11. The study groups AGEBs into Low, Medium, and High social lag categories but ignores the Very High and Very Low categories that might contain critical insights.

12. The omission of these categories raises concerns about selection bias in the analysis.

13. The study refers to flow-like landslides without precisely defining them in the Mexican geographical context. Are these debris flows, mudflows, or rock avalanches?

14. The classification of landslide types should be explicitly stated, preferably with references to recognized classification systems such as Cruden and Varnes (1996) or Hungr et al. (2014).

15. The study uses DEM data, land cover maps, and social indicators, but it does not explain:

i. The spatial resolution of all datasets and whether resampling was performed.

ii. How missing data was handled in the social lag variables.

iii. Whether the temporal mismatch between datasets (e.g., 2007 DEM, 2020 social data) could introduce bias.

16. The susceptibility model does not include uncertainty quantification. Given that landslide modeling is inherently probabilistic, the study should report confidence intervals for its estimates.

17. Several statements (e.g., social vulnerability theories, hydrological influence on landslides) lack citations from peer-reviewed sources.

Reviewer #2: A super simple straightforward application of FR for landslide hazard in a disorganized and unstructured draft.

English proficiency stands beyond quality due to frequent flaws, vague statements, linguistic flaws…

Improper references and commercial weblinks while solid publications are missing. Using a dissertation for FR from 2018 while this method has many solid references, ….

The next critical issue is the perquisite use of unified pixel size, and therefore it cannot be functional for extracted attributes from different imageries like Landsat, Modis, and … with different resolutions incorporated with raster data.

Therefore, based on the used datasets, it can only be acceptable when the inputs don’t change, else in the case of modified images (in pixel or grids) it will no longer match and hence, the integrity cannot be verified. In addition, the overlaying the raster and vector with different resolutions and geospatially distribution requires image enhance processing, or image fusion Technique. Novel approach can be found at https://www.sciencedirect.com/science/article/abs/pii/S0341816219303674, https://doi.org/10.1155/2023/9429505, …

Without any doubt, REJECTED conclusion. 1) pretty long and unjustified, 2) it obviously is not the place for citation

Overall,

1. Lack of novelty is concrete and beyond taking time for discussion

2. Doesn’t provide proper analyzed research gaps, taxonomy, and meaningful guidelines in the domain.

3. In comparison with advanced approaches For example, https://www.sciencedirect.com/science/article/abs/pii/S0341816219303674, https://link.springer.com/article/10.1007/s10346-019-01299-0, https://link.springer.com/article/10.1007/s100640050066, https://www.tandfonline.com/doi/abs/10.1080/02723646.2021.1978372, https://www.sciencedirect.com/science/article/pii/S2590056022000202, https://www.tandfonline.com/doi/abs/10.1080/01431161.2019.1672904, … it is uncompetitive.

4. Lack of data analysis/data visualization in considering the selected attributes are opaque

5. Lack of considering the effect of subsurface spatial soil/rock type distributions (https://link.springer.com/article/10.1007/s10064-018-1400-9, https://www.sciencedirect.com/science/article/abs/pii/S0169555X16306419, https://link.springer.com/article/10.1007/s10346-013-0409-1, https://www.sciencedirect.com/science/article/abs/pii/S0013795224002655, https://www.mdpi.com/2220-9964/10/5/341 ...), considering the clay sensitivity and its effect in landslide (https://www.sciencedirect.com/science/article/pii/S0037073817303019; https://www.sciencedirect.com/science/article/abs/pii/S0013795215000411, https://www.sciencedirect.com/science/article/pii/S0341816222002752 …), spatial analysis of the collected data and involved uncertainty of the post processing geo-model (https://link.springer.com/article/10.1007/s00366-023-01852-5, …

6. You have different attributes. The sensitivity analysis and predictability SHOULD be carried out via weight database of the optimum model to show the importance of each selected features (Search for updating the neural network models using different sensitivity analysis methods, sensitivity analysis for neural networks, novel feature selection using sensitivity analysis…)

7. Lack of any convincing and documented Discussion cannot be neglected. Solid comparison with other approaches and scholars/visualized results in compare with other scholars/computational time and cost based on the system and model used/technical limitations/uncertainty quantifications comparing with advanced automated predictive deep learning models..., impact of bias of the used data on the results, noise strategy removal, …

8. Trivially ill-formatted reference list

6. PLOS authors have the option to publish the peer review history of their article (what does this mean?). If published, this will include your full peer review and any attached files.

Reviewer #1: **Yes: **NAGARAJAN SANKARANARAYANAN

Reviewer #2: No

---

## [Author Response · Author response to Decision Letter 1]

18 Sep 2025

Dear Editor and Reviewers,

Thank you so very much for taking the time to make such a comprehensive review of our work. Thanks to it, we have come with a total refurbishment of the article: we have addressed each and every comment you kindly made, and also we made additions that go beyond the kind recommendations you made. We prepared a letter which address each point raised by the Editor and both of the reviewers plus the additions we deemed appropriate to fill the gaps you mentioned.

The revised Tracked changes version contains colored in Blue the new additions, and in yellow the parts which we took from the final version.

Editorial Comments.

The manuscript “Relationship between landslide susceptibility and social lag in Mexico City: The Case of the West Periphery” presents an interesting topic that integrates landslide susceptibility analysis with socio-economic factors. However, based on the two detailed reviews, there are significant methodological concerns that must be addressed before the manuscript can be reconsidered.

Key Issues to Address in the Revision:

1. Justification of the Frequency Ratio (FR) Method:

o Reviewer 1 suggests that the use of FR needs stronger justification, especially given the availability of more advanced machine learning techniques (e.g., Random Forest, SVM, CNN).

o A comparative analysis should be conducted between FR and an ML-based model to demonstrate its effectiveness.

Dear Editor and Reviewers, you are absolutely right, the justification presented to use the Frequency Ratio method is not enough. Consequently, we have now estimated 13 models to generate the susceptibility map, including but not limited to the ones suggested by the reviewers and editors e.g. Random Forest, SVM, Deep Learning and more. For each model we included the ROC curve and the AUC Value. This process allowed us to determine that Frequency Ratio was not the best model although it did a good job. We then changed Frequency Ratio by an XG Boost, which had the best performance according to the aforementioned metrics. The champion model was an XGBoost which is now used to generate the susceptibility map.

2. Weak Statistical Model for Social Lag Analysis:

o The multinomial logistic regression model shows an extremely weak fit (pseudo R² = 0.007352), making the relationship between social lag and landslide susceptibility statistically weak.

o Alternative modeling techniques (e.g., feature selection, interaction terms, or different regression models) should be considered.

Dear Editor, thank you so much for this comment. You are so right, we have now moved entirely to a new and different approach, using SHAP values derived from a Random Forest model to visually determine the association magnitude and direction. Additionally, we used this model to be able to calculate an R-squared metric and have a performance metric about how reliable this analysis is. We anticipate that we got 80% explained variance which is a decent statistic and which then provides credibility to our SHAP values. Furthermore, for each partition we made we estimated Copula Models to provide a robust magnitude and direction metric for the association between Landslide Susceptibility and Social Lag.

3. Validation of the Landslide Susceptibility Model:

o The lack of independent validation data is a major concern. It is crucial to include:

A validation dataset of past landslides to assess accuracy.

Performance metrics such as AUC-ROC curves to validate the predictive strength of the model.

You are absolutely right. In this new version, we include validation metrics for each susceptibility map created by 13 models. We include ROC and AUC and throughout them, it was possible to discriminate models, and to finally select the XGBoost.

4. Contradictions in Results and Hypothesis:

o The manuscript hypothesizes that higher social lag should correlate with higher landslide susceptibility, but findings show the opposite trend.

o A clearer explanation of this contradiction is required. If the hypothesis is incorrect, it should be revised with alternative theoretical support.

This revised version not only includes new models to create susceptibility maps, it also includes new ways to assess the relationship we are studying: Landslide Susceptibility vs Social Lag, in this new version we found significant relations between the study variables provided that we partition the dataset by deciles and using a Changepoint algorithm. Both partitions allowed us to study this relationship with more granularity, and it helped to avoid noise that comes from studying a large heterogeneous area such as the western periphery of Mexico City. With this new approach, it was possible to find a positive association in many deciles and segments partitioned by the Changepoint algorithm.

5. Choice of Huber Regression and Lack of Diagnostics:

o The rationale for using Huber regression should be justified over other robust regression techniques (e.g., Quantile Regression, Ridge Regression).

o The study lacks model diagnostics (e.g., residual analysis, VIF, or multicollinearity tests) to verify the robustness of Huber regression.

Again, you are absolutely right. In this new version we changed the models and we included performance metrics. As stated before, we first used random forest to derive SHAP Values as an exploratory way to visually study the magnitude and direction of the relationship, for this model we included R-squared metric for each of the segements. Secondly, we used Copulas as a robust association method and include the Loglikelihood coefficient.

6. Missing Data Issues and Temporal Bias in Data Sources:

o There is no explanation of how missing values in social lag variables were handled.

o The temporal mismatch between datasets (e.g., 2007 DEM vs. 2020 social data) could introduce bias.

Thank you very much for this suggestion. You are absolutely right, in this new version the whole Data Sources section was re-written and we included every relevant detail that is required, including the missing values, an explanation about the temporal mismatch -we anticipate that the dataset was not so mismatched, the maps used were 2017 not 2007 as we initially claimed, it was an error on our side and we apologize. Additionally, we include data about the re-projection and methods used, we are sure this new section is totally illustrative about data’s metadata.

7. Comparison with Official Landslide Hazard Maps:

o Given that Mexico City already has existing landslide hazard maps, the manuscript should compare its susceptibility map with official hazard maps to assess its reliability.

Thank you very much. This is obviously a valid concern. We found the official CENAPRED’s landslide susceptibility map. It is a categorized version which means it colors risk-zones according to a pre-defined range. Since this is the official version, we compared our susceptibility map with this one to provide a better assessment.

8. Lack of Uncertainty Quantification:

o Landslide susceptibility models are probabilistic in nature. The study should provide confidence intervals for its estimates.

We absolutely agree with this regard. To this end, we did the following “As a way to quantify uncertainty in the susceptibility map, we achieved this through a bootstrap resampling framework, which generates an ensemble of predictions to capture variability arising from sampling and model training processes. Specifically, we did 50 bootstrap iterations, wherein each iteration involves resampling the subsampled dataset with replacement to form a bootstrapped training set, followed by fitting an XGBoost classifier and predicting susceptibility probabilities across the cleaned feature matrix. The collection of these predictions, enables the derivation of ensemble statistics: the maximum (susceptibility_max) and minimum (susceptibility_min) values which represent the upper and lower bounds of prediction variability. These Max and Min maps make it straightforward to show the whole range of variability within the output.”

9. Inadequate Literature Citations and Reference Formatting Issues:

o Reviewer 2 highlights improper references, missing citations from high-impact peer-reviewed sources, and unstructured reference formatting.

o The literature review should incorporate recent and relevant citations, including key works on landslide susceptibility modeling.

Thank you very much for this comment. We revised our references and modified the main text and reference list accordingly; we added new ones and replaced the conflicting ones. The final version includes all of these changes.

10. Data Resolution and Spatial Analysis Issues:

• There is no explanation of how datasets with different spatial resolutions (e.g., Landsat, MODIS, and DEM data) were standardized.

• Image enhancement or fusion techniques may be required to ensure data consistency.

You are absolutely right. We re-write the vast majority of the Data Sources section to clarify exactly what we did about the spatial resolution, how it all fell into the same, comparable standard, the methods used, and the final resolution. We summarized all this information into a table at the end of this Data Sources section.

11. Missing Discussion:

• Sensitivity analysis should be performed to quantify the impact of different variables on landslide susceptibility.

• The manuscript lacks a proper discussion of computational efficiency, feature importance, and the limitations of the proposed method.

Thank you for this comment. In the new version we added three dimensions of sensitivity analysis: for the susceptibility map model we added XGBoost Feature Importance. Also, we added the Information Gain Ratio, and the GeoDetector q-statistic. For the association between SLI and Landslide susceptibility, in one setting, we calculated dependence between each of the variables with which SLI was made and Landslide Susceptibility.

Additionally, we included the limitations and efficiency in the XGBoost section since this was the model we used: “XGBoost complexity depends on the number of trees, features, tree-depth and data. It's parallel logic translates even complicated calculations into relatively simple threads. Although efficient, its limitations lie in that it has many hyperparameters to estimate, making it resource-intensive; it also tends to overfit data even with regularization; finally, it tends to be classified as black-box since the modeler is not fully aware of the model's inner calculations~\cite{bib72}”.

12. Both reviewers express concerns about language clarity, grammatical errors, and ambiguous phrasing. A thorough English language revision is necessary to ensure the manuscript is clear, coherent, and professionally structured.

I know, and I apologize about that! This version was reviewed several times, exclusively looking for grammar mistakes. We also tried to make sentences more concise.

###################################################################

Additional Editor Comments:

The manuscript “Relationship between landslide susceptibility and social lag in Mexico City: The Case of the West Periphery” presents an interesting topic that integrates landslide susceptibility analysis with socio-economic factors. However, based on the two detailed reviews, there are significant methodological concerns that must be addressed before the manuscript can be reconsidered.

Key Issues to Address in the Revision:

1. Justification of the Frequency Ratio (FR) Method:

o Reviewer 1 suggests that the use of FR needs stronger justification, especially given the availability of more advanced machine learning techniques (e.g., Random Forest, SVM, CNN).

o A comparative analysis should be conducted between FR and an ML-based model to demonstrate its effectiveness.

Sir, thank you so very much for your comment. We agree that the use of FR method needs stronger justification. Yet, we did something bigger: as suggested, we calculated several other models-12 in total- to generate the susceptibility map, including many Machine Learning models, and we compared them using the ROC/AUC plot and metric. The champion model was an XGBoost, and the justification for using it now is that it has the highest AUC.

2. Weak Statistical Model for Social Lag Analysis:

o The multinomial logistic regression model shows an extremely weak fit (pseudo R² = 0.007352), making the relationship between social lag and landslide susceptibility statistically weak.

o Alternative modeling techniques (e.g., feature selection, interaction terms, or different regression models) should be considered.

This is an absolutely valid concern. To completely solve it we recalculated everything by using new association methods which are SHAP values derived from a random forest, and COPULAS, which are excellent at modeling non-linear associations. Furthermore, to delve deeper into our dataset, we partitioned it into deciles and also by using a Changepoint partitioning algorithm. For each of the resulting segments, we applied the SHAP and COPULA Analysis resulting in many significant relationships. Additionally, to assess how valid was the application of the method, we used the r-squared derived from the random forest as a fitting metric. En each of the segments, that metric rounded 80%, which is a decent fit.

3. Validation of the Landslide Susceptibility Model:

o The lack of independent validation data is a major concern. It is crucial to include:

A validation dataset of past landslides to assess accuracy.

Performance metrics such as AUC-ROC curves to validate the predictive strength of the model.

Absolutely true, thank you for the comment. In this new version we included ROC-AUC plots and metrics for each of the models we used to create the landslide susceptibility map.

4. Contradictions in Results and Hypothesis:

o The manuscript hypothesizes that higher social lag should correlate with higher landslide susceptibility, but findings show the opposite trend.

o A clearer explanation of this contradiction is required. If the hypothesis is incorrect, it should be revised with alternative theoretical support.

Thank you so much again, and thanks for the patience of delving deeper into this point of our work. Yes, absolutely right, nevertheless we re-calculated the whole relationship with different methods and partitioning the dataset into deciles, with changepoints -and also doing it with the whole dataset. This new modeling approach did result into meaningful relationships which does not contradict the hypothesis. Thank you very much for the comment, and this limitation has been superseded.

5. Choice of Huber Regression and Lack of Diagnostics:

o The rationale for using Huber regression should be justified over other robust regression techniques (e.g., Quantile Regression, Ridge Regression).

o The study lacks model diagnostics (e.g., residual analysis, VIF, or multicollinearity tests) to verify the robustness of Huber regression.

Thank you for your valuable feedback. We agree that our initial use of Huber regression was a limitation, particularly regarding model diagnostics. To address this, we have replaced the Huber regression with a revised approach using SHAP values and Copula models. This change was made to enhance model interpretability and provide a more robust analysis. SHAP values directly address the lack of diagnostics by explaining the contribution of each feature to the model's output.

Regarding multicollinearity, we have maintained the exact covariates used by CONEVAL to ensure our findings are directly reproducible and comparable with the official Mexican Social Lag Index, which is a key objective of our study. This decision justifies including potentially collinear variables.

6. Missing Data Issues and Temporal Bias in Data Sources:

o There is no explanation of how missing values in social lag variables were handled.

o The temporal mismatch between datasets (e.g., 2007 DEM vs. 2020 social data) could introduce bias.

Thank you. Actually, it was a mistake on our side since the dataset dates 2017 except for the Land Cover and CONEVAL social lag, which dates 2020. Nevertheless, we address t

---

## [Decision Letter · Decision Letter 1]

18 Nov 2025

PONE-D-25-00532R1Relationship between landslide susceptibility and social lag in Mexico City: the Case of the West PeripheryPLOS ONE

Dear Dr. Mercado Mendoza,

Thank you for submitting your manuscript to PLOS ONE. After careful consideration, we feel that it has merit but does not fully meet PLOS ONE’s publication criteria as it currently stands. Therefore, we invite you to submit a revised version of the manuscript that addresses the points raised during the review process.

We look forward to receiving your revised manuscript.

Kind regards,

Gayathiri Ekambaram, Ph.D

Academic Editor

PLOS ONE

Journal Requirements:

Additional Editor Comments (if provided):

The updated manuscript covers a highly relevant issue, that is, combining machine learning-based landslide susceptibility maps with subtle studies of social underprivilege in a fast developing region. The peer reviews per the case indicate a significant disagreement in professional judgement. Reviewer 2 suggests rejection, the first major reason being insufficient and unsatisfactorily explained revisions, a continuing lack of English language fluency, a failure to respond systematically to comments made, unaddressed technical flaws (especially statistical analysis and data transparency), and no detailed responses, which are mapped to assigned line numbers. On the other hand, Reviewer 3 considers the changes in methodologies and presentation significant, applauding the use of more robust machine learning models (XGBoost, SHAP, Copula), a lot of uncertainty quantification, better literature integration, clarity in reporting on data handling, and succinctness of conclusions, and makes only limited changes to the text.

The revision is a step in the right direction as it provides considerable technical advancement, and the authors in an organized manner add complex models and enhance the robustness on the independent evaluation. The technical comments of Reviewer 3 have proper evidence of the novel version which has:

1. Comparison of the performance between different models (such as the XGBoost with an AUC of 0.883),

2. Time series bootstrap resampling,

3. Dependance diagnostics based on Copula and SHAP as alternatives to weak regressions,

4. Comprehensive and available methodological and data description, and

5. Better, but not perfect, English language and clarity.

Nevertheless, the fears expressed by Reviewer 2 with regards to the insufficiency of the revision response, quality of the language, the ongoing lack of clarity, and adherence to best practice in the documentation of response-to-review (the absence of a line numbering system and a clear mapping of comments to changes in the text) are legitimate. One can also notice the competitive positioning deficit with the latest state of the art, and some gaps in referencing and presentation of some of the results and figures.

Recommendation

Based on the differing reviews, it is evident that the manuscript has become better technically but is barely satisfactory with regards to the quality of communication and response formalism. This balance of evidence is not sufficient to allow outright rejection, as long as the deficiencies in remaining text and procedure are amended expressly.

Acceptance however should be subject to:

Response-to-review table submission Full mapping of all comments made by each prior reviewer/editor to certain line numbers and text modification in the revision,

One last pass of professional editing, preferably by a native or an identified editing agency,

Fixing of small textual problems indicated by Reviewer 3 (e.g., typos, figure captions, method/parameter transparency, and suggestion of future work).

In case of these conditions, it may be possible to proceed with the acceptance, the technical basis is now satisfactory and the work has methodological significance to policy and disaster risk mitigation in the urban peripheral setting.

Reviewers' comments:

Reviewer's Responses to Questions

**Comments to the Author**

1. If the authors have adequately addressed your comments raised in a previous round of review and you feel that this manuscript is now acceptable for publication, you may indicate that here to bypass the “Comments to the Author” section, enter your conflict of interest statement in the “Confidential to Editor” section, and submit your "Accept" recommendation.

Reviewer #2: (No Response)

Reviewer #3: All comments have been addressed

2. Is the manuscript technically sound, and do the data support the conclusions?

Reviewer #2: No

Reviewer #3: Yes

3. Has the statistical analysis been performed appropriately and rigorously?

Reviewer #2: No

Reviewer #3: Yes

4. Have the authors made all data underlying the findings in their manuscript fully available?

Reviewer #2: No

Reviewer #3: Yes

5. Is the manuscript presented in an intelligible fashion and written in standard English?

Reviewer #2: No

Reviewer #3: Yes

6. Review Comments to the Author

Reviewer #2: WONDERING. None of the commente neither are responded propoerly nor analytically discussed.

A simple evidence can be assigned to not revised Englsih as trivailly suffers from incohesion and inconssitent use of the third passive. Missing several comments, missing assigned line number corresponding to each comment or modification palce, pretty unjustified and long conclusion, ...

Reviewer #3: The authors have substantially revised the manuscript, addressing the major methodological concerns from the initial review. The integration of multiple machine learning models for susceptibility mapping, with XGBoost emerging as the top performer (AUC = 0.883), is a strong improvement. Validation through ROC/AUC curves, uncertainty quantification via bootstrap resampling, and comparison with the official CENAPRED map enhance the reliability of the results. The shift to SHAP values and Copula models for analyzing the relationship between landslide susceptibility and social lag provides a more nuanced, granular perspective, revealing regime-specific associations that were absent in the original version.

The literature review now includes recent, high-impact citations, and reference formatting has been corrected. Data handling details (e.g., resolutions, resampling, missing values) are clearly described. The discussion is more focused and interpretive, and the conclusion is concise.

The revisions resolve the key issues, making the manuscript suitable for publication with minor changes:

Correct the typo "socila lag" in the abstract to "social lag".

In the introduction (lines 81-82), rephrase for clarity: "as global warming intensifies and climatic events become more extreme, such as precipitation" to "as global warming intensifies extreme precipitation events".

Ensure all figure captions are descriptive and consistently formatted (e.g., Fig. 3: "Correlation matrix among environmental inputs used for susceptibility modeling").

In the methods, briefly note key XGBoost hyperparameters (e.g., n_estimators=100, learning_rate=0.1) for reproducibility.

Add a sentence to the conclusion suggesting future work, such as validating the model with real-time landslide data or incorporating climate projections.

I recommend acceptance after these minor revisions.

7. PLOS authors have the option to publish the peer review history of their article (what does this mean?). If published, this will include your full peer review and any attached files.

Reviewer #2: No

Reviewer #3: No

---

## [Author Response · Author response to Decision Letter 2]

9 Dec 2025

Dear Editor and Reviewers,

Thank you very much for your kind comments and useful reviews in the last round. In what follows, we will provide a detailed in-line numbered review. We highlighted in green sections containing line-numbers, figure numbers and pages so that it will be easier to locate changes. We separated the response in two sections. The first one (current document) addresses the issues raised in the first round of comments, and the second addresses the issues regarding the second round, mainly referring to language issues and minor changes.

We sincerely hope this revised version reflects the transparency and quality work you are looking for.

Section 1.

Rebuttal letter to the first set of comments.

The revised Tracked changes version contains colored in Blue the new additions, and in yellow the parts which we took from the final version. Furthermore, in this revised rebuttal letter we highlighted in green the additions which in essence are the association from each suggestion to the line number in which the change was performed. Unless specifically stated, this set of comments and replies is associated with the First and Original tracked changes document which is labeled as______________

Editorial Comments.

The manuscript “Relationship between landslide susceptibility and social lag in Mexico City: The Case of the West Periphery” presents an interesting topic that integrates landslide susceptibility analysis with socio-economic factors. However, based on the two detailed reviews, there are significant methodological concerns that must be addressed before the manuscript can be reconsidered.

Key Issues to Address in the Revision:

1. Justification of the Frequency Ratio (FR) Method:

o Reviewer 1 suggests that the use of FR needs stronger justification, especially given the availability of more advanced machine learning techniques (e.g., Random Forest, SVM, CNN).

o A comparative analysis should be conducted between FR and an ML-based model to demonstrate its effectiveness.

Dear Editor and Reviewers, you are absolutely right, the justification presented to use the Frequency Ratio method is not enough. Consequently, we have now estimated 13 models to generate the susceptibility map, including but not limited to the ones suggested by the reviewers and editors e.g. Random Forest, SVM, Deep Learning and more. We present the models in page 8 beginning at line 312 all the way through line 392. This initial presentation refers to a general model introduction. Because of lack of space, Latex Editor moved a summary table below the section which includes each model‘s most important characteristics located in page 10, and since it is a Latex element it only used one numbered line, after line 418. Furthermore, model results which are the susceptibility maps are located in page 22 Figure 5 after line 802. In addition, the Champion model is depicted alone in Page 22 Figure 6 after line 809.

For each model we included the ROC curve and the AUC Value. A table containing these results is located in page 22 after line 797 and the Receiver Operating Curve (ROC) Plots are located in page 21 after line 791. This process allowed us to determine that Frequency Ratio was not the best model although it did a good job. We then changed Frequency Ratio by an XG Boost, which had the best performance according to the aforementioned metrics. The champion model was an XGBoost which is now used to generate the susceptibility map, as just mentioned, it is located on page 22 Figure 6 line 809.

2. Weak Statistical Model for Social Lag Analysis:

o The multinomial logistic regression model shows an extremely weak fit (pseudo R² = 0.007352), making the relationship between social lag and landslide susceptibility statistically weak.

o Alternative modeling techniques (e.g., feature selection, interaction terms, or different regression models) should be considered.

Dear Editor, thank you so much for this comment. You are so right, we have now moved entirely to a new and different approach, using SHAP values derived from a Random Forest model to visually determine the association magnitude and direction. We briefly describe SHAP values and the beeswarm plots in page 13 lines 486 - 498. In addition, SHAP plots are located in three sections: Full Dataset SHAP plots which, along with the corresponding explanatory text, are located in lines 957-986; CHangepoints section in lines 987-1038; and Decile section in lines 1040-1070.

Additionally, we used this model (A Random Forest) to calculate the R-squared metric. We anticipate that we got 80% R-squared, implying that, our analysis explained variance which is a decent statistic and which then provides credibility to our SHAP values. For Full dataset located in lines 958-962; Changepoints 992-997; and Decile separation in lines 1040-1048

Furthermore, for each partition we made we estimated Copula Models to provide a robust magnitude and direction metric for the association between Landslide Susceptibility and Social Lag.

Copula Theory can be found in lines 499-537. Estimations are separated in three sections which are Full dataset, changepoint induced partition and deciles. For the Full dataset, changes are in lines 974-986; for changepoint induced partitions changes are located in lines 1020-1038 ; and for decile partitions changes are in lines 1062-1070

3. Validation of the Landslide Susceptibility Model:

o The lack of independent validation data is a major concern. It is crucial to include:

A validation dataset of past landslides to assess accuracy.

Performance metrics such as AUC-ROC curves to validate the predictive strength of the model.

You are absolutely right. In this new version, we include validation metrics for each susceptibility map created by 13 models. We include ROC and AUC and throughout them, it was possible to discriminate models, and to finally select the XGBoost. These changes are incorporated in lines 393-408, which corresponds to the meaning of these variables, and the numerical/graphical implementations can be found in lines 787-802, there can be found a summary table containing AUC values as well as the actuall Matrix ROC plot for every implemented model. Furthermore, with respect to the independent validation for which a validation dataset is to be added, we included into the inputs repository the original World Bank Median from 1980-2018 about rainfall triggered landslide map (Median). The complete repository is open, free and fully accessible in Zenodo: https://doi.org/10.5281/zenodo.17156313

4. Contradictions in Results and Hypothesis:

o The manuscript hypothesizes that higher social lag should correlate with higher landslide susceptibility, but findings show the opposite trend.

o A clearer explanation of this contradiction is required. If the hypothesis is incorrect, it should be revised with alternative theoretical support.

Thank you very much for the kind comment. This new version changed completely the approach since the new models (Copula and SHAP plots derived from Random Forest) now have agreeable goodness of fit values (GoF) of R-squared around 0.8, and using these models with decent GoF there is no contradiction. This revised version not only includes new models to create susceptibility maps, it also includes new ways to assess the relationship we are studying: Landslide Susceptibility vs Social Lag, in this new version we found significant relations between the study variables provided that we partition the dataset by deciles and using a Changepoint algorithm. Both partitions allowed us to study this relationship with more granularity, and it helped to avoid noise that comes from studying a large heterogeneous area such as the western periphery of Mexico City. With this new approach, it was possible to find a positive association in many deciles and segments partitioned by the Changepoint algorithm. New association results can be found in lines 918-1070, including the following variants:

Full dataset: lines 995-986

Changepoint induced partitions: lines 987-1038

Deciles: lines 1039-1070

Individual association between landslide susceptibility and each of the variables with which the social lag index was built in line 1173.

5. Choice of Huber Regression and Lack of Diagnostics:

o The rationale for using Huber regression should be justified over other robust regression techniques (e.g., Quantile Regression, Ridge Regression).

o The study lacks model diagnostics (e.g., residual analysis, VIF, or multicollinearity tests) to verify the robustness of Huber regression.

Again, you are absolutely right. In this new version we changed the models and we included performance metrics. As stated before, we first used random forest to derive SHAP Values as an exploratory way to visually study the magnitude and direction of the relationship, for this model we included R-squared metric for each of the segements. Secondly, we used Copulas as a robust association method and include the Loglikelihood coefficient. As with the previous answer, we totally agree with the reviewers, which is why we totally dumped the Huber / Robust regression approach to the aforementioned SHAP and Copula approaches. New association results can be found in lines 918-1070, including the following variants:

Full dataset: lines 995-986

Changepoint induced partitions: lines 987-1038

Deciles: lines 1039-1070

Individual association between landslide susceptibility and each of the variables with which the social lag index was built in line 1173.

6. Missing Data Issues and Temporal Bias in Data Sources:

o There is no explanation of how missing values in social lag variables were handled.

o The temporal mismatch between datasets (e.g., 2007 DEM vs. 2020 social data) could introduce bias.

Thank you very much for this suggestion. You are absolutely right, in this new version the whole Data Sources section was re-written and we included every relevant detail that is required, including the missing values, an explanation about the temporal mismatch -we anticipate that the dataset was not so mismatched, the maps used were 2017 not 2007 as we initially claimed, it was an error on our side and we apologize. Additionally, we include data about the re-projection and methods used, we are sure this new section is totally illustrative about data’s metadata. To begin with, detailed data presentation begins at line 632 all the way through line 755. Specifically, an initial data presentation is located in lines 632-678. A Tau DEM presentation is in lines 679-755. We have now added a summary table with the input name, resolution, transformation and data source.line 751. Particularly, missing data is treated in lines 643, 670-676. Furthermore, the mismatch, which as we stated above, was a confusion on our side, is clarified in lines 640-641, and lines 649-654.

7. Comparison with Official Landslide Hazard Maps:

o Given that Mexico City already has existing landslide hazard maps, the manuscript should compare its susceptibility map with official hazard maps to assess its reliability.

Thank you very much. This is obviously a valid concern. We found the official CENAPRED’s landslide susceptibility map. It is a categorized version which means it colors risk-zones according to a pre-defined range. Since this is the official version, we compared our susceptibility map with this one to provide a better assessment. The resulting comparison between our generated categorized susceptibility map begins with the categorized susceptibility map in line 826 and the next paragraph which explains the similarity between lines 826-837. Furthermore, and more importantly, CENAPRED official susceptibility map is in line 837 Figure 9.

8. Lack of Uncertainty Quantification:

o Landslide susceptibility models are probabilistic in nature. The study should provide confidence intervals for its estimates.

We absolutely agree with this regard. To this end, we did the following “As a way to quantify uncertainty in the susceptibility map, we achieved this through a bootstrap resampling framework, which generates an ensemble of predictions to capture variability arising from sampling and model training processes. Specifically, we did 50 bootstrap iterations, wherein each iteration involves resampling the subsampled dataset with replacement to form a bootstrapped training set, followed by fitting an XGBoost classifier and predicting susceptibility probabilities across the cleaned feature matrix. The collection of these predictions, enables the derivation of ensemble statistics: the maximum (susceptibility_max) and minimum (susceptibility_min) values which represent the upper and lower bounds of prediction variability. These Max and Min maps make it straightforward to show the whole range of variability within the output.” This information is located between lines 810-819 with the associated plot which is in line 820 (Figure 7 Maximum and Minimum plots)

9. Inadequate Literature Citations and Reference Formatting Issues:

o Reviewer 2 highlights improper references, missing citations from high-impact peer-reviewed sources, and unstructured reference formatting.

o The literature review should incorporate recent and relevant citations, including key works on landslide susceptibility modeling.

Thank you very much for this comment. We revised our references and modified the main text and reference list accordingly; we added new ones and replaced the conflicting ones. The final version includes all of these changes. References are located in lines 1532 onwards. Noteice that Latex Editor overlapped two tables inside the references, but this will be edited and fixed in the final version.

10. Data Resolution and Spatial Analysis Issues:

• There is no explanation of how datasets with different spatial resolutions (e.g., Landsat, MODIS, and DEM data) were standardized.

• Image enhancement or fusion techniques may be required to ensure data consistency.

You are absolutely right. We re-write the vast majority of the Data Sources section to clarify exactly what we did about the spatial resolution, how it all fell into the same, comparable standard, the methods used, and the final resolution. We summarized all this information into a table at the end of this Data Sources section. An expanded explanation lies between lines 633-755. Specifically, in lines 636 we specify the reference coordinate system; line 649 specifies which images were not altered given they were at the same resolution. Lines 647-652 shows the transformations applied to this input; Lines 657-662 shows the transformation uses for WB data, as well as in lines 673-676. A summary is provided in lines 737-745 and Table 4 provides a summary which includes Resolution, transformation and source. This table is licated after line 751

11. Missing Discussion:

• Sensitivity analysis should be performed to quantify the impact of different variables on landslide susceptibility.

• The manuscript lacks a proper discussion of computational efficiency, feature importance, and the limitations of the proposed method.

Thank you for this comment. In the new version we added three dimensions of sensitivity analysis: for the susceptibility map model we added XGBoost Feature Importance. Also, we added the Information Gain Ratio, and the GeoDetector q-statistic. For the association between SLI and Landslide susceptibility, in one setting, we calculated dependence between each of the variables with which SLI was made and Landslide Susceptibility. The results are in plot after line 857, and the plot´s interpretation are between lines 838-857. As just stated, the techniques used are Information Gain Ratio, Feature Importance, and the q-statistic

Additionally, we included the limitations and efficiency in the XGBoost section since this was the model we used: “XGBoost complexity depends on the number of trees, features, tree-depth and data. It's parallel logic translates even complicated calculations into relatively simple threads. Although efficient, its limitations lie in that it has many hyperparameters to estimate, making it resource-intensive; it also tends to overfit data even with regularization; finally, it tends to be cla

---

## [Editor Report · Decision Letter 2]

14 Dec 2025

PONE-D-25-00532R2

Relationship between landslide susceptibility and social lag in Mexico City: the Case of the West Periphery  Dear Dr. Mercado Mendoza , Thank you for submitting your manuscript to PLOS ONE. After careful consideration, we feel that it has merit but does not fully meet PLOS ONE’s publication criteria as it currently stands. Therefore, we invite you to submit a revised version of the manuscript that addresses the points raised during the review process. Please submit your revised manuscript by Jan 28 2026 11:59PM. If you will need more time than this to complete your revisions, please reply to this message or contact the journal office at plosone@plos.org. Please include the following items when submitting your revised manuscript:

We look forward to receiving your revised manuscript.

Kind regards,

Gayathiri Ekambaram, Ph.D

Academic Editor

PLOS One

Additional Editor comments:

The Revised Manuscripts have substantially improved the methodological rigor and clarity of the work. However, kindly address the minor issues:

Although language has improved, some sentences remain long or awkward and would benefit from one more round of careful English editing for clarity and conciseness.Ensure that all tables and figures are properly placed and numbered in the final layout (some overlap issues are noted in the LaTeX output and references section).Check internal consistency of line/figure/table cross‑references after typesetting and remove any residual references to earlier versions of the manuscript (e.g., “first version,” “this new version”).

---

## [Author Response · Author response to Decision Letter 3]

19 Dec 2025

Kindest Sirs,

We are deeply thankful for your quick and kind reply. We have prepared a list with numbered lines with respect to each of the suggested changes. Again, we are deeply thankful for the efforts you’ve made to improve so much the quality of our work.

Although language has improved, some sentences remain long or awkward and would benefit from one more round of careful English editing for clarity and conciseness.

Thank you so very much for your kind comment. The whole article was completely revised, and to resolve the issues associated with this comment, the following changes were implemented:

Grammatical issues.

Abstract: we corrected earthflow to earthflow.

Lines 303-304. “The methods here presented” / to / “The methods presented here”.

Lines 513-514. “they allow to model” / to / “they allow researchers to model”.

Lines 530-531. “Table 3 engulfs” / to / “Table 3 summarizes”.

Lines 538-539. “we did’t used GPU’s” / to / “we didn’t use GPUs”.

Line 698 . “every tested models” / to / “every model tested”

Lines 776-777. “within the each AGEB” / to / “within each AGEB”.

Line 791. “Susceptibilitytends” / to / “Susceptibility tends”.

Line 454. “Find -if there exist- a relationship” / to / “find whether there exists a relationship”.

Lines 624-625. “the D-Infinity algorithm, offers on one hand, an advanced approach ” / to / “the D-Infinity algorithm, offers an advanced approach”.

Long paragraphs that were separated and shortened

Lines 30-37. Paragraph made concise.

Lines 46-51. Paragraph split and shortened.

Lines 62-68. Paragraph shortened.

Lines 69-71. Paragraph shortened.

Lines 102-110. Paragraph split.

Lines 123-130. Paragraph split.

Lines 137-141. Paragraph shortened.

Lines 141-152. Paragraph split.

Lines 198-201. Paragraph modified for conciseness.

Lines 232-242. Paragraph split.

Lines 603-608. Paragraph shortened.

Lines 611-620. Paragraph shortened.

Lines 784-798. Paragraph changed for consistency.

Lines 855-863. Paragraph split.

Lines 1101-1115. Paragraph split.

Lines 1125-1137. Paragraph split.

Ensure that all tables and figures are properly placed and numbered in the final layout (some overlap issues are noted in the LaTeX output and references section).

Thank you for your kind comment. The final layout was revised to fulfill PLOS ONE norms on figures, tables, and numbering. Particularly, we introduced several pagebreaks in Latex to avoid overlapping between bullets, tables, figures, and text. Moreover, the next particular changes were performed:

Figures were modified in the following aspects:

Figures 1 through 13 incorporated a revised boldfaced title.

Figures 1 through 13 incorporated a short description.

Figures 5 through 13 were moved next to the first time they are called in the text.

Added cross-references within the text for figures 5-10

Tables were modified as follows:

Table 1. Table notes added.

Table 2. Added the Table notes legend.

Table 3. Added Table notes and removed parts of equations to make formulas fit.

Table 4. Note modified.

Table 6. Note modified.

Furthermore, to correct the specific issue mentioned about the overlapping with references, at the end of the document, we inserted page breaks to separate the final sections (Supporting information and Acknowledgements) from the Appendix, and from References in lines 1215 onwards. The result is a now a non-overlapping document.

Check internal consistency of line/figure/table cross‑references after typesetting and remove any residual references to earlier versions of the manuscript (e.g., “first version,” “this new version”).

Thank you very much for this kind comment. We didn´t notice before but there where many Tables and Figures that were not cross-referenced. Furthermore, some of those that were referenced, the cross-reference was placed improperly in the sense that, it was more than two paragraphs away from either the Figure or Table that was being called. All of these errors are now corrected in the final layout. Specifically, we added crossreference for figures and Tables 5-onwards.

---

## [Editor Report · Decision Letter 3]

26 Dec 2025

Dear Authour,

Congratulations on a comprehensive revision that significantly strengthens this valuable socio-environmental risk study.

Kind regards,

Gayathiri Ekambaram, Ph.D

Academic Editor

PLOS One

---

## [Editor Report · Acceptance letter]

PONE-D-25-00532R3

PLOS One

Dear Dr. Mercado Mendoza,

I'm pleased to inform you that your manuscript has been deemed suitable for publication in PLOS One. Congratulations! Your manuscript is now being handed over to our production team.

Kind regards,

on behalf of

Dr. Gayathiri Ekambaram

Academic Editor

PLOS One